# LLMs Are In-Context Bandit Reinforcement Learners

**Giovanni Monea,**[1] **Antoine Bosselut,**[2] **Kianté Brantley,**[3] **and Yoav Artzi**[1]
[1]Cornell University     [2]EPFL     [3]Harvard University
`{giovanni, yoav}@cs.cornell.edu, antoine.bosselut@epfl.ch,`
`kdbrantley@g.harvard.edu`

## Abstract

Large Language Models (LLMs) excel at in-context learning (ICL), a supervised learning technique that relies on adding annotated examples to the model context. We investigate a contextual bandit version of in-context reinforcement learning (ICRL), where models learn in-context, online, from external reward, instead of supervised data. We show that LLMs effectively demonstrate such learning, and provide a detailed study of the phenomena, experimenting with challenging classification tasks and models of sizes from 500M to 70B parameters. This includes identifying and addressing the instability of the process, demonstrating learning with both semantic and abstract labels, and showing scaling trends. Our findings highlight ICRL capabilities in LLMs, while also underscoring fundamental limitations in their implicit reasoning about errors.

## 1 Introduction

Large language models (LLMs) have been shown to exhibit in-context learning (ICL), a form of supervised learning that does not require parameter updates (Brown et al., 2020). ICL relies on including supervised input-output pairs in the LLM context (i.e., prompt),[1] and it has proven effective with few (Brown et al., 2020) and many (Bertsch et al., 2024; Agarwal et al., 2024) examples. We ask whether the ability to learn in-context extends to contextual bandit reinforcement learning (RL), i.e., whether language models can effectively perform in-context reinforcement learning (ICRL) with stateful single-step interaction episodes.

ICRL naturally combines ICL and reinforcement learning (RL). In contrast to ICL, as an RL process, ICRL does not rely on annotated labels or a fixed dataset. Instead of constructing the LLM context from supervised input-output pairs, the LLM context is built from triplets of an input, a model's predicted output, and its reward. As more input examples are observed, the model has access to additional triplets in context, leading to an online and continual learning scenario, where model capabilities improve over time. These triplets are followed by a

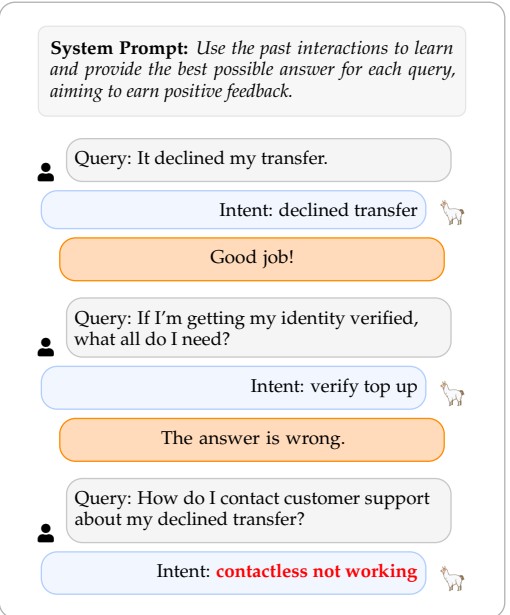

Figure 1: **Illustration of in-context bandit reinforcement learning.** The context shows a sequence of user queries , model responses , and feedback in the Banking77 77-label classification domain. The model learns in-context from rewards given to its previous predictions. The final prediction (shown in red) represents the model's current guess.

---

[1]We use the terms *prompt* and *context* interchangeably.

new input, for which the model must predict an output. In this in-context framework, adding a past episode to the context corresponds, in standard fine-tuning RL settings, to using an episode at training time. Figure 1 illustrates bandit ICRL prompting.

ICRL is a desirable ability of LLMs. It allows learning new tasks interactively, in an online setting, at deployment time (without parameter updates), without requiring demonstration data. This learning signal may be human-generated, automatically provided (i.e., a program successfully completes; Gehring et al., 2024), or even AI-generated (Zhang et al., 2024a).

We study the bandit ICRL capabilities of Llama 3.1 (Llama Team, 2024), Qwen2.5 (Qwen et al., 2024), and Gemini 1.5 Flash (Gemini Team, 2024).[2] Following existing bandit learning literature (Zhang et al., 2019; Bietti et al., 2021), we use many-label classification benchmarks to create contextual bandit RL scenarios, which simplify experimentation and evaluation, while focusing on the fundamental skills of exploration and learning from rewards.

We find that LLMs demonstrate innate ICRL capabilities, but that two choices are critical for effective learning. First, a relatively high stochasticity level is needed to encourage exploration. Second, using only triples with positive rewards performs best.The latter choice creates a cosmetic similarity to ICL. However, ICRL remains fundamentally different: in ICRL the model must actively explore to find (i.e., generate) these positive triplets, rather than having an expert annotator provide them.

A recurring observation in our experiments is the relative instability of the process, as performance can suddenly dip significantly, before often quickly recovering. We propose a new method, *Stochastic ICRL*, to add stochasticity to the prompt construction by only sampling some of the past episodes observed in context, instead of increasing the temperature. This enhances exploration, stabilizes performance, and maintains relatively shorter contexts. Interestingly, this also allows the model to learn in the presence of negative signals.We also study the scaling trends of ICRL, using all Qwen2.5 modeling sizes between 0.5B and 72B. There is a strong correlation between performance and model scale, but across all scales the relation between methods is maintained, with regard to both performance and stability.

Overall, we demonstrate that applying bandit ICRL consistently and significantly enhances the performance of LLMs. For example, in the Banking77 (Casanueva et al., 2020) classification task, Qwen2.5-7B improves from 6.2% zero-shot accuracy to 72.2% through Stochastic ICRL, without access to gold labels and without any updates of model parameters. These results suggest that LLM hold previously understudied capabilities for in-context learning, and lay the foundation for their further development and study in future work. Our code, data, and experimental logs are available at `https://lil-lab.github.io/icrl/`.

## 2 In-context Reinforcement Learning

ICL (Brown et al., 2020) operates by providing a model with annotated demonstrations of a task. A demonstration includes an input (e.g., *What is the best football club in Europe?*) and its corresponding annotated output (e.g., *AC Milan*). ICL's reliance on pre-existing gold-standard labeled data follows the common supervised learning paradigm, although without any change in the model parameters.

ICRL follows the reinforcement learning paradigm (Sutton & Barto, 2018), where models learn by reinforcing their own good behaviors and suppressing their own bad choices. Instead of providing models with correct demonstrations, the model generates an output given an input, then observe the outcome (i.e., reward) of its prediction. It learns from the reward signals, in an online learning setting, all within the context (i.e., without parameter updates). In this study we focus on a contextual bandit RL scenario, a restricted variant of RL, where the length of each episode is one step.

Formally, the model $\pi$ observes an input $x^{(t)} \sim \mathcal{D}$ sampled from the data distribution $\mathcal{D}$ at time $t$, generates a prediction $\hat{y}^{(t)}$, and then observes a reward $r^{(t)} \sim R(x^{(t)}, \hat{y}^{(t)})$. We

---

[2]We also conduct early experiments on Phi-3.5-mini (Abdin et al., 2024), included in the appendix.

**Algorithm 1** Naive and Naive+ ICRL

**Require:**
    $\mathcal{D}$: Data distribution
    $\pi$: Language model policy
    $R$: Reward function

1: Init buffer $\mathcal{E} \leftarrow \varnothing$
2: **for** $t = 1, 2, 3, \ldots$ **do**
3:     $C \leftarrow \mathcal{E}$
4:     Observe input $x^{(t)} \sim \mathcal{D}$
5:     Sample prediction $\hat{y}^{(t)} \sim \pi(\cdot \mid C, x^{(t)})$
6:     Observe reward $r^{(t)} \sim R(x^{(t)}, \hat{y}^{(t)})$
7:     **if** $r^{(t)} \leq 0$ **then**    ⎫  Only in
8:        **Continue** to next $t$ ⎬  Naive+ ICRL
9:     **end if**     ⎭
10:   Add episode to buffer $\mathcal{E} \leftarrow \mathcal{E} \cup \{(x^{(t)}, \hat{y}^{(t)}, r^{(t)})\}$
11: **end for**

**Algorithm 2** Stochastic ICRL

**Require:**
    $\mathcal{D}$: Data distribution
    $\pi$: Language model policy
    $R$: Reward function
    $p_{\text{keep}}$: Prob. to keep examples in context

1: Init episode buffer $\mathcal{E} \leftarrow \varnothing$
2: **for** $t = 1, 2, 3, \ldots$ **do**
3:     Init empty context $C^{(t)} \leftarrow [\,]$
4:     **for** $e \in \mathcal{E}$ **do**
5:        $b \sim \text{Bernoulli}(p_{\text{keep}})$
6:        **if** $b = 1$ **then**
7:          Add episode to context $C^{(t)} \mathrel{+}= e$
8:        **end if**
9:     **end for**
10:   Observe input $x^{(t)} \sim \mathcal{D}$
11:   Sample prediction $\hat{y}^{(t)} \sim \pi(\cdot | C^{(t)}, x^{(t)})$
12:   Observe reward $r^{(t)} \sim R(x^{(t)}, \hat{y}^{(t)})$
13:   **if** $r^{(t)} > 0$ **then**
14:     Add episode to buffer
        $\mathcal{E} \leftarrow \mathcal{E} \cup \{(x^{(t)}, \hat{y}^{(t)}, r^{(t)})\}$
15:   **end if**
16: **end for**

denote the tuple $(x^{(t)}, \hat{y}^{(t)}, r^{(t)})$ as an episode. This formulation does not assume access to datasets of correct demonstrations, but to a reward (i.e., feedback) function.

In common RL terminology, the model $\pi$ is the policy, the input $x^{(t)}$ is the state,[3] and the prediction $y^{(t)}$ is the action. Throughout our formulation, the policy is also conditioned on previous episodes in the form of an LLM context, similar to how supervised examples are provided in ICL. These past episodes are not part of the RL state. Instead, the context is used to perform in-context policy improvement, similar to how past episodes are used to perform policy improvement in conventional RL (e.g., via parameter updates).

We design several methods to elicit ICRL from LLMs. The Naive approach is a straightforward implementation of ICRL following the common ICL recipe (Section 2.1). The Stochastic approach (Section 2.2) is an alternative to increasing sampling temperature, but with more stability. In Appendix B.3, we propose Approximate ICRL, an additional approach that shares similarities with Stochastic, while being more efficient in high-memory setups.

## 2.1 Naive and Naive+ ICRL

Algorithm 1 describes the Naive approach, as well as Naive+, a variant that only considers examples with positive reward. Omitting lines 7–8 gives Naive ICRL, the most straightforward way to implement ICRL. At each time step $t$, the model observes a new example $x^{(t)}$, produces a prediction $\hat{y}^{(t)}$, and receives a reward $r^{(t)}$. Every such model interaction creates an episode, which is appended to the buffer $\mathcal{E}$. For each new interaction, we construct a context $C$ from prior episodes (line 3). In Naive, this context is simply all past episodes. Naive+ ICRL adds lines 7–8 and ignores negative-reward episodes, retaining only positive episodes in the buffer. As the LLM's context window fills, both variants maintain a sliding window by dropping older episodes.

Empirically, Naive does not learn effectively (Section 4; Figure 2), while Naive+ is very effective, especially with relatively high sampling temperature. The gap between the two indicates the failure of Naive is due to the presence of examples with negative reward.

---

[3]In the bandit literature, the state is often called *context*, and hence the name contextual bandits. We do not use this term to avoid confusion with the LLM context.

Critically, even when only using positive examples, ICRL still differs from supervised ICL in that it relies on the model's generations rather than a fixed set of annotated demonstrations. This much more challenging scenario necessitates the model to explore and iteratively refine its outputs through reinforcement; without this capability, further learning does not occur.

## 2.2 Stochastic ICRL

Stochastic ICRL utilizes model sensitivity to prompt composition as an avenue to increase exploration, instead of the increased temperature of Naive. Changes in prompt composition have been widely observed to lead to variance in LLM behavior, including through changes in the set of ICL examples (Zhang et al., 2022; Liu et al., 2022; Chen et al., 2023; Levy et al., 2023), seemingly meaningless stylistic changes (Sclar et al., 2024; Lu et al., 2022), or even interventions based on entropy in the embedding space (Rahn et al., 2024). Generally, this property of LLMs is not viewed positively. However, it adds stochasticity to the ICRL process, which encourages exploration.

Stochastic introduces context stochasticity by randomly choosing the subset of past episodes to include in the prompt for each new input. Like Naive+ ICRL, it includes only positive-reward episodes, which improves results empirically.

Algorithm 2 describes Stochastic ICRL. For each input, we construct a new context (lines 3–9). We decide what past episodes to include in this context by sampling from a Bernoulli variable parameterized by $p_{\text{keep}}$ (lines 4–9). We sample independently for each past episode. This results in different implicit reasoning for each input, because each is done with a different context. As in Naive+, we only store episodes with positive reward (lines 13–15).

With small $p_{\text{keep}}$, Stochastic will encounter the issue of the LLM context window saturating much later than Naive. However, deploy ICRL for enough interactions, and the context window will saturate, even for models with the largest windows.

Similar to naive, we downsample the context if it overflows the LLM context window. We do it by removing episodes from the sampled $C^{(t)}$ uniformly at random until the context fits the model's context window.

## 3 Experimental Setup

**Models**   We use the instruction-tuned versions of Llama 3.1 8B (Llama Team, 2024) and Qwen2.5 (Qwen et al., 2024) for all model sizes.[4] For the hardest tasks, we also experiment with Gemini 1.5 Flash 8B (Gemini Team, 2024).[5] We use all models in a multi-turn chat format. We compute the maximum number of episodes the context window can take for each model-task combination (Appendix C.2). We use constrained decoding to generate model predictions, as in recent work on ICL (Bertsch et al., 2024).

**Tasks**   We follow Bertsch et al.'s (2024) study of many-shot ICL in focusing on five classification problems: Banking77 (77 labels; Casanueva et al., 2020), CLINC150 (150 labels; Larson et al., 2019, NLU (68 labels; Liu et al., 2021), TREC (6 labels; Li & Roth, 2002; Hovy et al., 2001), and TREC-fine (50 labels; Li & Roth, 2002; Hovy et al., 2001). Because of the large output spaces (up to 150 labels in CLINC150), these tasks are particularly challenging for large language models, as empirically shown by Bertsch et al. (2024) and replicated in our zero-shot results. The classification problem creates a contextual bandit scenario (Zhang et al., 2019; Bietti et al., 2021). The labels in each dataset are used to compute rewards, and are never shown to the model. Appendix C.3 offers more details on the datasets.

The datasets are of different sizes. The size of the datasets dictates the number of time steps in our experiments. We randomly sub-sample Banking77, CLINC150, and NLU to

---

[4]We include in the appendix early experiments with Phi-3.5-mini (Abdin et al., 2024), which generally performs worse due to relative model weakness.

[5]We limits our experiments with Gemini due to costs. Overall, we spent $2,120 USD on Gemini API calls.

10k examples. TREC and TREC-fine are smaller, so we only use 5k training examples for each. This allows the experiments to be of relatively standard length. The training data corresponds to the data distribution $\mathcal{D}$ in our algorithms. We also sub-sample all test sets to 500 examples each, to reduce the computational cost of experiments. NLU does not provide a standard test set, so we create our own train and test splits. In all experiments, the datasets contain the same examples in the same order.

**Semantic vs. Abstract Class Names**   We study using both the original class names and abstract labels. The original class names carry important *semantic* information, which gives the model helpful clues on how input examples map to them (e.g., the output class name *calendar update* in CLINC150 is a strong hint to which input queries may apply to it). Experiments with abstract labels remove this information by mapping all labels to meaningless abstract strings (e.g., *label5*). Experiments and results use the original (semantic) labels by default, unless noted explicitly that they are using abstract labels.

**Rewards and Prompt Design**   We simulate interactive binary rewards from perfect automatic verifiers or human actors interacting with the system. We do so by comparing the model outputs with the gold label of each input. This is a common practice in studies of the effect of rewards on learning processes (Gao et al., 2023; Lightman et al., 2024), for practical convenience. We deterministically transform the binary numerical rewards into a natural language format indicating if the model prediction is correct or not, which is more suitable for LLM reasoning. This formulation abstracts over challenges like exact numerical interpretation (i.e., of continuous rewards), while focusing on the fundamental skills of exploration and learning from rewards. Appendix C.1 elaborates on our prompting.

**Evaluation**   We report running test accuracy, using the held-out test set of each dataset. We compute it every 500 steps for each test example separately, using the context used to process that step's training example. In the appendix, we also report regret, the forgone utility from an actual model prediction in comparison to the oracle choice.

**Comparisons**   We compare ICRL with the zero-shot setting, which corresponds to the performance on the test set without any in-context examples.[6] We also report a supervised ICL upper bound by testing performance with the maximum possible number of gold-standard supervised demonstrations in context. These results use expert demonstrations, which the ICRL results have no access to in our learning process. As expected these ICL results outperform the ICRL trends we report. However, the reliance on annotated demonstrations makes them not comparable to the ICRL results. We provide them to get an idea of the upper bound of ICL in these scenarios.

## 4   Results and Analysis

Figure 2 shows the test accuracies for Llama 3.1 8B and Qwen2.5 7B. As an upper bound to in-context learning, we also show the performance of an oracle with access to the gold labels for the maximum number of in-context examples that the model can fit. Unless specified otherwise, we use $p_{keep} = 0.1$ and sampling temperature $T = 1.0$ for Stochastic and zero-shot, $T = 1.0$ for Naive, and $T = 2.0$ for Naive+. We choose the best parameters for each ICRL method and include our analysis in Appendix B.

**LLMs Learn In-Context From Their Own Predictions and Rewards**   Both Naive+ and Stochastic effectively learn in all tasks and for both models, showing significant improvements over zero-shot. Naive+ and Stochastic improve over the zero-shot accuracies by between 28.6–74.4% for Llama, and 29.2–68.4% for Qwen. In general, accuracies approach

---

[6]We visually highlight the zero-shot performance with sampling temperature $T = 1.0$, which is the temperature we use for Naive and Stochastic experiments. Zero-shot accuracy with Naive+'s temperature of 2.0 can still be observed by looking at the accuracy at the first step of the Naive+ curves.

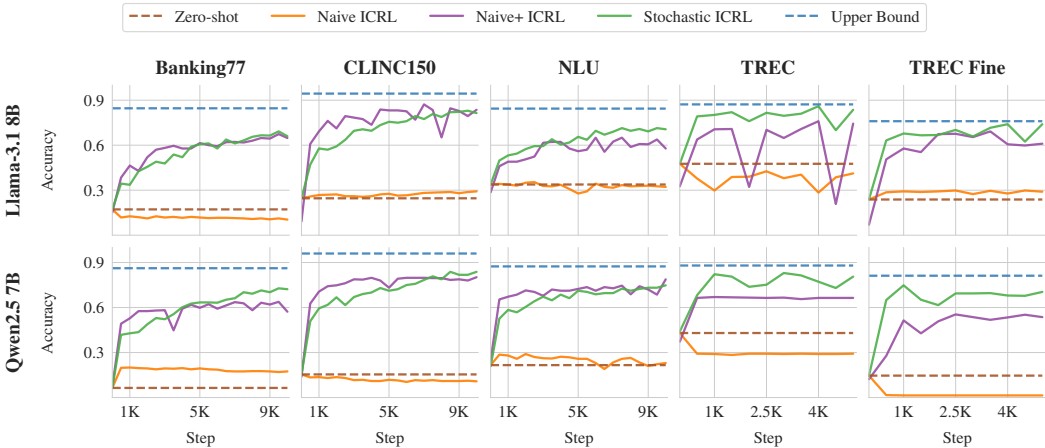

Figure 2: **Performance of ICRL.** Naive, Naive+, and Stochastic held-out test results for Llama and Qwen and all tasks. Naive+ and Stochastic consistently outperform zero-shot (i.e., first step) and Naive, while also showing consistent trends of continual improvement as more data is observed. Table 2 in Appendix D details start and end accuracies.

the supervised performance in many settings, demonstrating the strong bandit ICRL capabilities of Llama and Qwen. Performance also grows monotonically over time, especially with Stochastic, suggesting further gains with more data. This trend is most evident in the most challenging datasets (Banking77, CLINC150, NLU), where high label counts demand more exploration to map inputs to outputs.

**Reward Signals Are Crucial, but Mistakes Remain Challenging** Figure 3 shows ablations studies. Removing rewards or inverting them brings about negligible gains over zero-shot performance for both Naive and Stochastic models. This demonstrates that learning is driven by the reward signals, and not simply by the inclusion of domain examples in the context (i.e., domain effect; Min et al., 2022; Pan et al., 2023; Lyu et al., 2023; Kossen et al., 2024).

Unlike Naive, which only performs effectively when exclusively positive-reward episodes are considered (i.e., Naive+), Stochastic partially maintains its learning capabilities even when exposed to negative outcomes. Although its performance is negatively impacted, this suggests that our stochastic approach prevents the model from becoming overwhelmed by signals it struggles to interpret. Notably, Stochastic remains robust even when 10% of the rewards are inverted (i.e., noisy), indicating resilience to noise, which is likely in human-feedback settings.

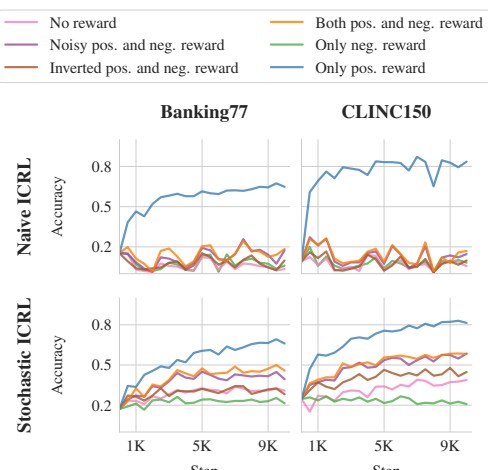

Figure 3: **Reward ablations.** Test accuracies of Naive and Stochastic with different reward signals. Positive reward only is the best choice for both methods. With Naive, no other strategy facilitates learning. Table 3 in Appendix D details start and end accuracies.

Overall, our ablations show that (a) LLMs can learn online from their predictions only when a reward signal is present, and (b) LLMs exhibit inherent limitations in implicitly learning from mistakes (i.e., without explicit reasoning, as in Wei et al.'s (2022) *Chain-of-Thought*).

**Label Semantics Contribute, but ICRL Occurs Without It** Previous supervised ICL work has shown that LLMs can learn tasks whose labels have no semantic meaning (Pan et al.,

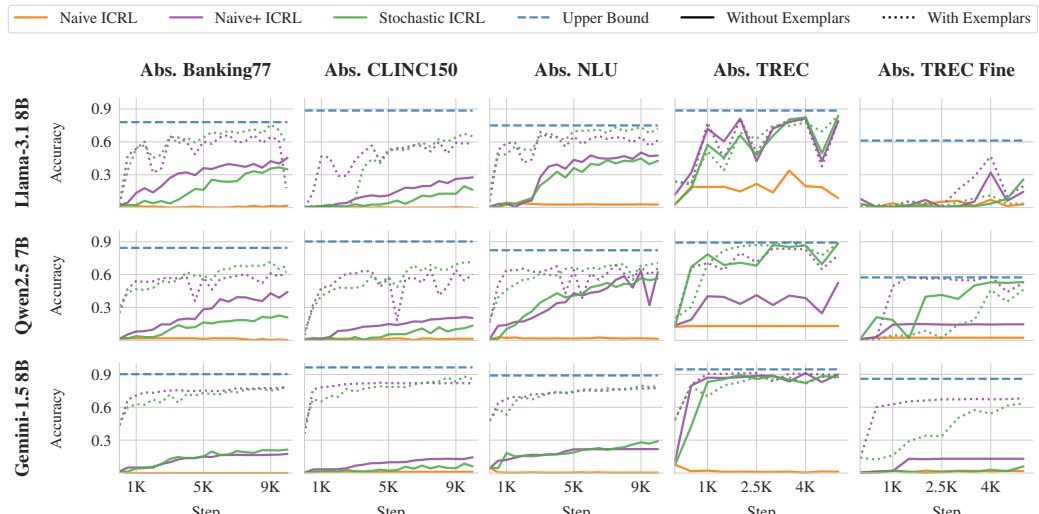

Figure 4: **ICRL with Abstract Labels.** We evaluate whether LLMs can learn tasks whose labels carry no semantic meaning by mapping each label to label_{number}. Even without initial exemplar demonstrations, Qwen and Llama show increasing performance over time. Gemini similarly excels when given an initial mapping, but struggles in a purely exploratory setting. Table 4 in Appendix D details start and end accuracies.

2023; Li et al., 2024), that is, tasks with *abstract labels*. This poses a harder challenge than with labels with semantic meaning, because cannot rely on pre-trained input-output associations. We experiment with removing all semantic information from the label space, by mapping each original label to a format label_{number}.[7] This ensures that the labels themselves carry no meaningful information that might help the model. We evaluate two scenarios. In the first and more challenging setup (*Without Exemplars*), we provide no prior demonstrations of correct input-output mappings, thus adhering closely to the ICRL protocol. In the second setup (*With Exemplars*), we give exactly one correct demonstration per label at the start of the prompt (i.e., before past episodes).

Figure 4 summarizes our findings on Llama 8B, Qwen2.5 7B, and Gemini 1.5 Flash 8B. To contextualize the results, we include upper bound results from standard supervised ICL with as many gold demonstrations (with abstract labels) as the context can handle, which generally succeeds across tasks (except for a lower performance on TREC-fine). With just one exemplar per label, ICRL nearly reaches the upper bound for all tasks and models. We also stress that just including one gold demonstration per label is not always enough to reach good performance, and the online process is still important: for example, Llama with exemplars, when tested on CLINC150 with Stochastic, reaches non-negligible accuracies only after 3k steps. In the absence of any exemplars, the ICRL process still manages to build informative contexts, though overall accuracy is not surprisingly lower. For instance, Qwen and Llama achieve higher than 45% accuracy on NLU and TREC with both Naive+ and Stochastic, indicating that the models learn and refine their output mappings over time. In contrast, Gemini excels when the correct mapping is provided but struggles significantly in a purely exploratory setting.

Given the domain effect observed with semantic labels, and the relatively low, even if significant results observed with abstract labels, the question arises if this learning is due to a domain effect. For example, Llama on CLINC150 improves from 0.8→27.6% during learning with Naive+, a significant, but modest improvement. We ran additional experiments to study the presence of learning with abstract labels, but without reward signals (i.e., to measure

---

[7]We assign each unique label a random integer from 1000 onward, up to the total number of labels in a given task.

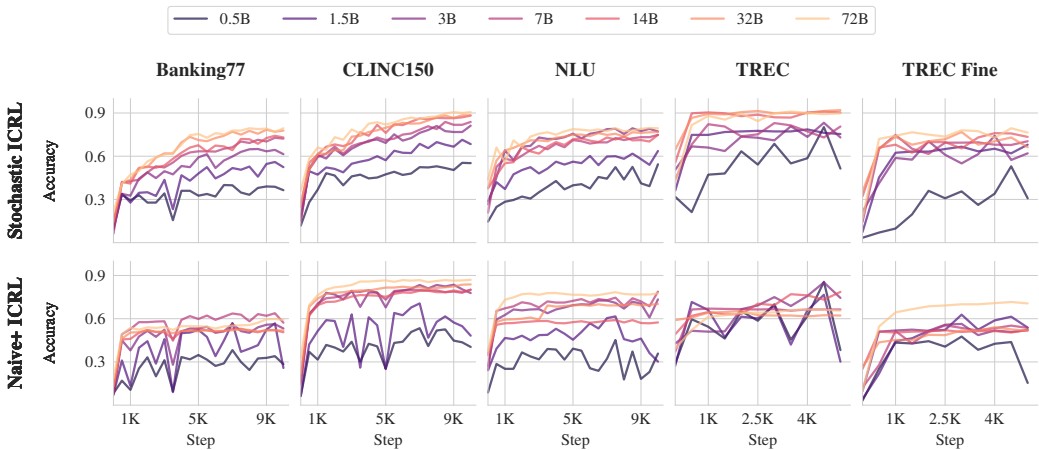

(a) **ICRL Scaling Trends.** We evaluate Qwen models from 500M to 70B parameters using both Naive+ and Stochastic. Performance improves substantially over zero-shot accuracy for all sizes, but smaller models plateau at lower accuracies, indicating that ICRL performance correlates positively with model size. Table 5 in Appendix D details start and end accuracies.

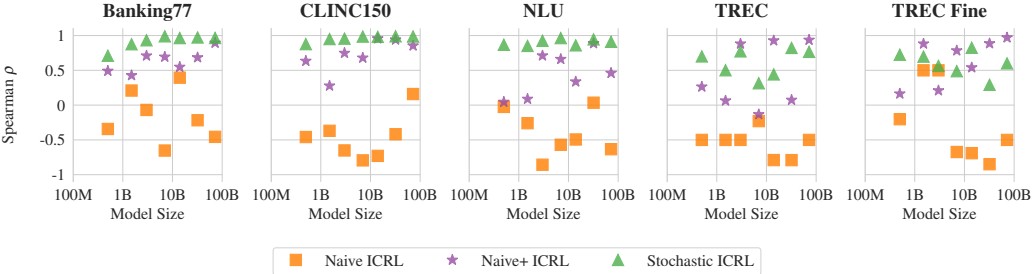

(b) **Stability of Naive+ and Stochastic ICRL.** We measure stability by computing Spearman's rank correlation between accuracy and time step. Except for TREC and TREC-fine (which give inconclusive results), Stochastic exhibits more stable learning on Banking77, CLINC150, and NLU. Larger models also show higher stability, mirroring trends in (a). Table 6 in Appendix D details $\rho$ values.

Figure 5: **Comparison of Qwen models (500M–72B).** We analyze scaling accuracy gains (a) and stability differences (b).

the domain effect). We see no improvement over zero-shot performance, confirming the reward signal is what drives learning.[8]

**Bigger Models Are Better at ICRL**  We evaluate Qwen Instruct models ranging from 500M to 72B parameters using both Naive+ and Stochastic to characterize the scaling trends of ICRL. Figure 5a shows the results. For all model sizes, performance improves substantially over zero-shot accuracy (measured at the first time step). However, smaller models tend to plateau at lower accuracies compared to larger models, indicating that ICRL benefits from model scale, similar to other LLM behaviors.

**Stochastic is More Stable Than Naive+**  An important differentiating factor between Stochastic and Naive+ is stability. The results so far (Figures 2, 4, and 5a) often show sudden, even if temporary dips in performance with Naive+. This instability is undesirable, because it means the performance of the model in interactions (i.e., with users) temporarily deteriorates significantly. In contrast, Stochastic's learning trends are more stable.

---

[8]Because these experiments showed no learning effect at all, we are omitting them from our figures.

We quantify stability as Spearman's rank correlation ($\rho$) between accuracy and time step.[9] Figure 5b shows the relation between stability and model sizes for all Qwen models for the three methods. Except for TREC and TREC-fine, which give inconclusive results, Stochastic exhibits more stable learning than Naive+ on Banking77, CLINC150, and NLU across all model scales. As expected, in general, larger models show higher stability, mirroring the trends in Figure 5a. We hypothesize that Stochastic is less sensitive to short-term fluctuations, as it relies on a smaller but more diverse set of episodes at each step.

## 5 Related Work

**Supervised ICL**   ICL was first demonstrated by Brown et al. (2020), and since then its causes (Chan et al., 2022; Xie et al., 2022; Olsson et al., 2022; Garg et al., 2022; Von Oswald et al., 2023; Hendel et al., 2023; Wang et al., 2023) and the level of learning it displays (Min et al., 2022; Lyu et al., 2023) have been studied extensively. By now, it is well established that LLMs can learn new tasks in context (Garg et al., 2022; Wei et al., 2023; Pan et al., 2023; Kossen et al., 2024; Li et al., 2024). Our work builds on this line of work, and provides the first evidence that LLMs have the innate capability to perform RL in context, and not only supervised learning (i.e., the standard way it is done), in the contextual bandit setting.

Our study would not be possible without recent increases in the context window length of LLMs (Llama Team, 2024; Abdin et al., 2024; Gemini Team, 2024). Recent work showed that model performance can continue to increase when including hundreds or thousands of ICL demonstrations (Bertsch et al., 2024; Agarwal et al., 2024). We find similar results: LLMs can continually improve when learning through ICRL until their context does not saturate. Interestingly, while some work (Zhang et al., 2024b; Mo et al., 2024; Shinn et al., 2023) find that models can learn from mistakes, we do not observe effective learning from episodes with negative rewards. It is possible that models can learn from mistakes only when explicitly reasoning (Kojima et al., 2022; Wei et al., 2022) about them (Shinn et al., 2023; Zhang et al., 2024b), but not implicitly. This is an important direction for further study.

**ICRL**   Likely the closest work to ours is Krishnamurthy et al.'s (2024) study of whether LLMs can solve multi-armed bandit problems, a state-less simpler RL setting than the one we are focused on. We observe similar issues to their findings with the Naive approach. They present a set of negative results, and finally are able to elicit effective learning, but through a prompting strategy that cannot generalize beyond their very simple scenario. We address this challenge by showing the strong performances of both Naive+, which includes only positive outcomes and an increased sampling temperature, and Stochastic, which features stochasticity in the prompt construction. Concurrent to our work, Nie et al. (2024) studied the contextual bandit ICRL, similar to our study. They also propose a working solution, but take a very different approach by externally tracking learning statistics commonly used in the UCB bandit learning algorithm (Auer et al., 2002), and fine-tuning or prompting a model to leverage them. In contrast, the approaches we discuss do not rely on explicitly tracking statistics or fine-tuning. Instead, we are interested in innate abilities.

Wu et al. (2024) propose benchmarks that include a simplified multi-armed bandit problem. Their baseline results with a method similar to Naive show mixed results in a setting that is even simpler than that of Krishnamurthy et al. (2024).

A few studies have also considered multi-step RL toy settings (Brooks et al., 2023; Mirchandani et al., 2023). Mirchandani et al. (2023) prompt models to improve past trajectories, and Brooks et al. (2023) simulate policy iteration with LLMs. Both works find that models cannot learn in general. Interestingly, Mirchandani et al. (2023) attribute this failure to LLMs inability to explore and find optimal solutions, as we also observed in our analysis of Naive.

**Transformers and RL**   Another related line of research is that of Transformers trained to solve sequential decision-making problems (Janner et al., 2021; Chen et al., 2021; Xu et al.,

---

[9]A perfect positive correlation ($\rho = 1$) indicates strictly increasing accuracy over time, whereas $\rho = -1$ means performance strictly decreases.

2022; Laskin et al., 2022; Zheng et al., 2022; Lee et al., 2023; Grigsby et al., 2024; Raparthy et al., 2024). In all these cases, Transformers (Vaswani et al., 2017) are trained from scratch. Our focus is different: we study ICRL that emerges from the process of training LLMs, without fine-tuning the LLM for this purpose.

## 6    Discussion and Limitations

We study the innate capabilities of off-the-shelf LLMs to perform ICRL in the contextual bandit setting. We outline a straightforward algorithm to show this behavior, and propose an enhanced version featuring stochasticity in the prompt construction, while increasing stability. We characterize ICRL, including scaling effects, stability, the importance of the reward signal, and the impact of abstract labels (i.e., that contain no semantic information).

Fundamentally, our work illustrates that exploration is the key ingredient necessary for ICRL behavior in LLMs. When exploration is combined with filtering of episodes with negative rewards, conventional ICL abilities (i.e., learning from demonstrations) bring about strong ICRL trends. Furthermore, exploration can be aided by introducing stochasticity in the prompt construction. The dependence of the learning trends on filtering out negative rewards leaves an important challenge for future work – how to elicit or train LLMs to reason effectively about negative episodes.

While our work provides a plethora of insights into ICRL behavior, much remains to be studied. We intentionally choose the contextual bandit setting using classification benchmarks following (Zhang et al., 2019; Bietti et al., 2021), and focus on binary rewards to simplify the experiments and evaluation in this early stage of studying ICRL. This formulation abstracts over challenges like exact numerical interpretation (i.e., of rewards), while focusing on the fundamental skills of exploration and learning from rewards. However, this limitation leaves open the question of applicability to more complex RL problems, where rewards are more nuanced, or where interactions comprise multiple steps. For example, math and coding tasks often require multiple steps, but also introduce complex evaluation challenges. We believe our study enables future work to study these challenges, and that this is an important direction.

Our work also leaves open questions about the use of computational resources. ICRL is relatively compute-intensive, especially after the learner observes many episodes. We propose Approximate ICRL in Appendix B.3 to reduce certain forms of computational overhead, and show how it allows to trade-off compute for robustness. Further reducing computational demands is an important direction for future work.

We hope our work helps to shed light on the capabilities of contemporary LLMs, and that it lays out the ground for extensive future work, both in research and practice.

## Acknowledgments

We thank Yair Feldman for proposing Spearman's rank correlation as a stability metric, and Mustafa Omer Gul, Yair Feldman, Yilun Hua, and Robert West for insightful discussion and feedback. This research was supported by NSF under grants No. 1750499 and OAC-2311521, NASA under award No. 20-OSTFL20-0053, a gift from Open Philanthropy, the Academic Grant Program, a gift from Apple, the National Artificial Intelligence Research Resource (NAIRR) Pilot, the Frontera supercomputer supported by the National Science Foundation (award NSF-OAC 1818253) at the Texas Advanced Computing Center (TACC) at The University of Texas at Austin, and the Delta advanced computing and data resource which is supported by the National Science Foundation (award NSF-OAC 2005572). We thank Google for enabling experiments with Gemini through a gift. We gratefully acknowledge use of the research computing resources of the Empire AI Consortium, Inc, with support from the State of New York, the Simons Foundation, and the Secunda Family Foundation (Bloom et al., 2025). AB gratefully acknowledges the support of the Swiss National Science Foundation (No. 215390), Innosuisse (PFFS-21-29), the EPFL Center for Imaging, Sony Group Corporation, and the Allen Institute for AI. Any opinions, findings and con-

clusions or recommendations expressed in this material are those of the author(s) and do not necessarily reflect the views of the National Science Foundation, NASA, or the other funders.

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

## A  Evaluation Measures in the Appendix

In the appendix, in multiple cases, we also report regret, the forgone utility from an actual model prediction in comparison to the oracle choice. Intuitively, regret measures how many interactions the model handled poorly throughout the experiment. In our experiments, regret is the accumulated number of incorrect examples throughout learning. Regret gives a single number that considers both the final performance and how fast the model reached it. A good system would reach high performance as fast as possible, making fewer mistakes overall (i.e., would have a low regret).

In some cases, we also report train accuracy as the running mean accuracy over the most recent 256 episodes.

## B  Additional Method Analysis

### B.1  Naive and Naive+ ICRL Hyperparameters

We study the effect of the model sampling temperature $T$ on both Naive and Naive+ ICRL (Figure 6). For Naive (Figure 6a), we observe that varying $T$ does not significantly affect performance, and all values lead to relatively poor results. In contrast, Naive+ ICRL is highly sensitive to $T$ (Figure 6b): while higher temperatures can sometimes reach stronger performance, they also introduce substantial instability. Low temperatures are more stable but plateau at lower levels of accuracy. Overall, we find that $T = 2.0$ achieves both good performance and stability. We adopt this value for all subsequent Naive+ experiments, including the ablations reported in Figure 3.

A related concern involves zero-shot performance (i.e., performance at time step 0). Because we use $T = 1.0$ for Stochastic (and Naive) and $T = 2.0$ for Naive+, it is unclear whether to measure zero-shot performance with $T = 1.0$ or $T = 2.0$. To ensure fairness, in all experiments combining Naive+ and Stochastic, we report the higher of these two zero-shot accuracies as our baseline. In particular, in many instances, because of this choice, the difference between final and initial performance exceeds the difference between final and zero-shot performance.

### B.2  Stochastic ICRL

#### B.2.1  Downsampling Strategies

In our formulation of Stochastic ICRL, we downsample too large contexts by randomly removing selected episodes until they fit the model context. However, we design three strategies in total to downsample the context if we reach the limit of the LLM context window: (a) *unbiased* (the default strategy): randomly remove episodes from $C^{(t)}$ until it fits the context window; (b) *start-biased*: use the longest possible prefix of episodes from $C^{(t)}$ such that it fits the LLM context size; and (c) *end-biased*: use the longest possible suffix. Unbiased corresponds to the approach used in the main paper.

In practice, we never saturate the LLM context window when using Stochastic ICRL with $p_{\text{keep}} = 0.1$ because our context windows are more than 100k. We conduct experiments to evaluate the above strategiesby limiting the context window of Llama to 4k or 8k tokens. Generally, we observe that *start-biased* strategy outperforms *unbiased*, which in turn performs better than *end-biased*, in all cases, although by only small margins. Given these results, we focus on *unbiased* as the most straightforward approach. Figure 8 shows the results of this analysis for Banking77 and CLINC150.

#### B.2.2  Hyperparameter Tuning and Sensitivity

Stochasticity in context generation is one of the key components that contribute to both Stochastic performance. It is controlled by setting $p_{\text{keep}}$. Figure 7 shows the sensitivity of Stochastic to different values of $p_{\text{keep}}$. Without stochasticity ($p_{\text{keep}} = 1.0$), ICRL struggles

on both models—particularly on Phi —while setting $p_{\text{keep}}$ too low retains too few examples in the context and hurts performance. Setting $p_{\text{keep}} = 0.1$ strikes a good balance, yielding strong results while keeping the context short (and therefore faster to run). We fix $p_{\text{keep}}$ to 0.1 for all subsequent Stochastic experiments.

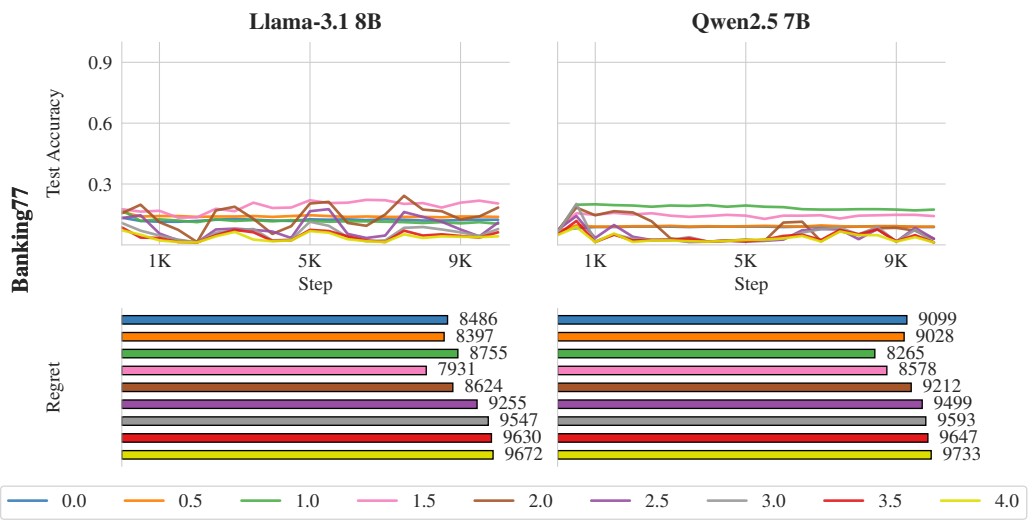

(a) **Temperature Sensitivity Analysis for Naive.**

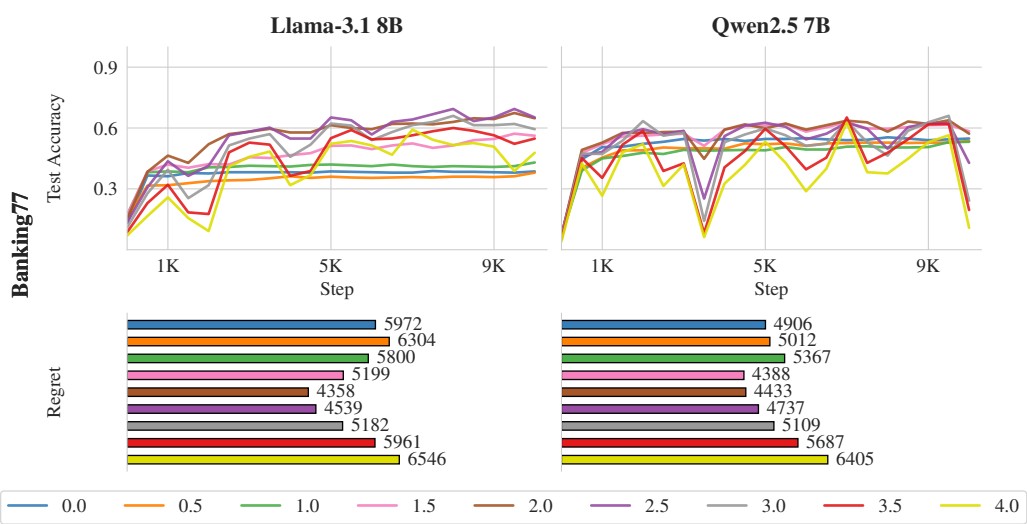

(b) **Temperature Sensitivity Analysis for Naive+.**

Figure 6: **Temperature Sensitivity Analysis for Naive and Naive+.** We plot the performance of each approach across different sampling temperatures $T$. (a) For Naive, varying $T$ has little impact, and all temperature settings result in relatively poor performance. (b) Naive+ exhibits significant variability: higher $T$ values can lead to strong performance but are less stable, whereas lower $T$ values yield more stable results but with lower peak accuracy. We choose $T = 2.0$ for Naive+ to achieve both strong performance and stability.

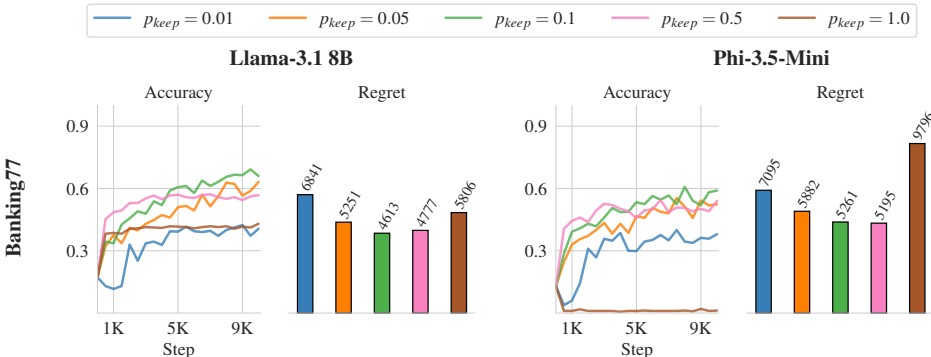

Figure 7: **Sensitivity to $p_{\mathbf{keep}}$ in Stochastic ICRL**. We compare performance with different values of $p_{\text{keep}}$. Intermediate values learn better for both Llama and Phi.

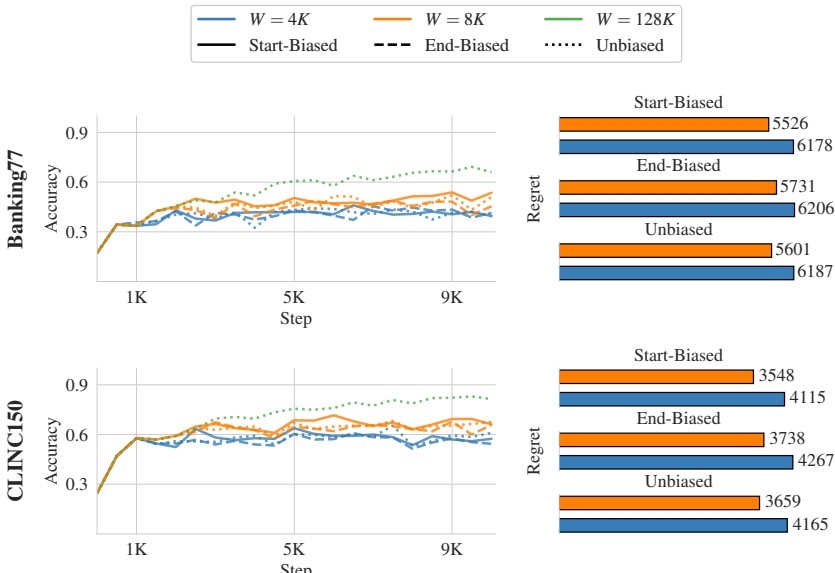

Figure 8: **Varying Maximum Context in Llama for Banking77 and CLINC150.** Comparison of test accuracy and regret of Llama under varying context lengths and subsampling strategies for both Banking77 and CLINC150 datasets. Longer contexts generally enhance performance, with subtle differences observed between subsampling strategies. The difference between the strategies is negligible.

We do not optimize the sampling temperature $T$ for Stochastic in this work and simply fix it to a standard value of 1.0. It is possible that performance could improve further with a more optimal temperature selection. We leave this investigation to future work.

## B.3 Approximate ICRL

### B.3.1 Stochastic ICRL Computational Costs

An important technical difference between the Naive and Naive+ approaches and Stochastic is that, until the context window is not saturated, Naive approaches can potentially re-use past computations from caching. This is not possible in Stochastic, because each episode requires the construction of a fresh context $C^{(t)}$. The probability of encountering the same

---

**Algorithm 3** Approximate ICRL

---

**Require:**
    Everything from Algorithm 2
    $K$: Number of contexts to maintain

1: Init empty contexts $\mathcal{C} \leftarrow \{[]^{(1)}, \ldots, []^{(K)}\}$
2: **for** $t = 1, 2, 3, \ldots$ **do**
3:     Sample context uniformally $C \sim \mathcal{U}(\mathcal{C})$
4:     Observe input $x^{(t)} \sim \mathcal{D}$
5:     Sample prediction $\hat{y}^{(t)} \sim \pi(\cdot | C, x^{(t)})$
6:     Observe reward $r^{(t)} \sim R(x^{(t)}, \hat{y}^{(t)})$
7:     **if** $r > 0$ **then**
8:         **for** $k = 1$ to $K$ **do**
9:             $b \sim \text{Bernoulli}(p_{\text{keep}})$
10:            **if** $b = 1$ **then**
11:                Add episode to cached context
$$\mathcal{C}[k] \mathrel{+}= (x^{(t)}, \hat{y}^{(t)}, r^{(t)})$$
12:            **end if**
13:        **end for**
14:    **end if**
15: **end for**

---

context twice, or even the same prefix, is exceptionally low even after a few episodes. This means that the context has to be computed from scratch for each input.[10]

### B.3.2   *Method*

We propose an approximation of Stochastic ICRL that balances between computational cost and learning effectiveness. Similar to both Naive+ and Stochastic, the approximate version also excludes episodes with negative reward and, like Stochastic, focuses on exploration by stochasticity in the context.

Algorithm 3 describes Approximate ICRL. The core idea behind the approximation is to persistently store a limited number of contexts, so we can simply gradually expand them with new episodes, rather than always create and compute new contexts. We maintain $K$ contexts $\mathcal{C}$, which all start empty (line 1). At each time step $t$, we sample a context $C$ from the $K$ contexts (line 3), and use it for episode $t$ (lines 4–6. If the reward $r^{(t)} > 0$, we use the episode to expand all contexts stochastically. For each context in $\mathcal{C}$, we expand it with the $t$-th episode with a probability of $p_{\text{keep}}$ (lines 8–11).

Approximate introduces stochasticity in two places: sampling the context to use for each episode and the expansion of the stored contexts. In Algorithm 3, we use *uniform* sampling to choose the context (line 3). This is a uniform approximation of the probability of a context, which can also be easily computed *exactly* using the probabilities of the episodes it contains and $p_{\text{keep}}$. In practice, we find the exact computation to work poorly, because contexts that are assigned more episodes or have low probability episodes quickly receive very low probability, and are not used. Figure 10b shows this experimental analysis. We use uniform sampling throughout our experiments.

The level of approximation the algorithm provides depends on the resources available. For example, one can allocate each context to a compute unit, so a machine with eight compute units (e.g., GPUs) will support $K = 8$. Approximate is a strict approximation of Stochastic in the sense that coupling the exact context sampling strategy with $K \to \infty$ gives Stochastic. However, the approximation is limited in handling contexts that extend beyond the LLM

---

[10]In low-memory setups, this does not lead to noticeable slowdowns (as efficient caching would not be possible), and Stochastic can be much faster given that each context contains only $p_{\text{keep}}\%$ of the episodes that Naive+ would use. In our setup, we empirically find this to be the case and Stochastic is significantly faster in practice.

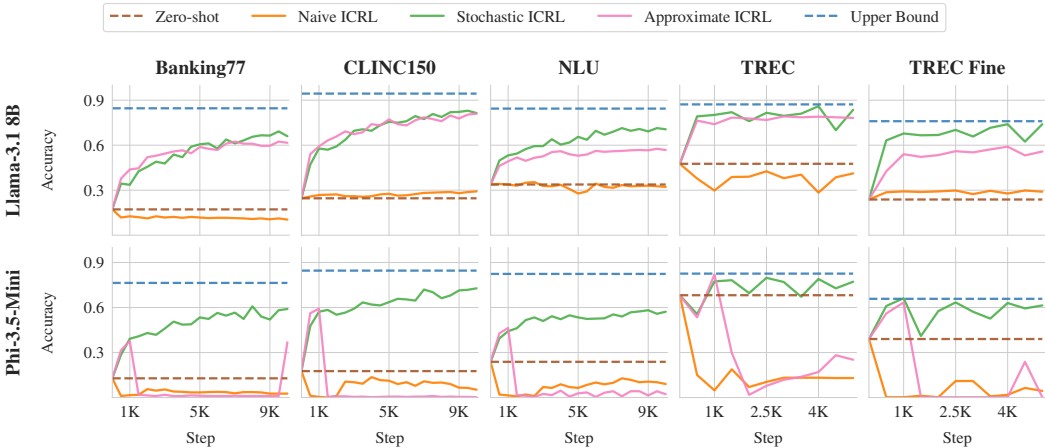

Figure 9: **Performance of ICRL**. Stochastic and Approximate held-out test results for both Llama and Phi and all semantic-labels tasks.

window length. Overcoming this while maintaining the efficiency of the approximation is an important direction for future work.

### B.3.3 Results

We test Approximate on Llama and Phi only, and show the results in Figure 9. If not specified, we choose $K = 8$ for Approximate.

**Approximate is an Effective Alternative to Stochastic** In Figure 9, Approximate performs almost as well as Stochastic ICRL when using Llama, across all tasks. The results are very different with Phi: despite early learning, Approximate deteriorates quickly. This stems from one of the contexts being biased towards one label and therefore predicting only this label. Eventually, the bias towards the label spreads to other contexts, leading to the collapse in performance we observe. It is empirically possible to recover, as we see in Banking77 later in the experiment, but the chance of it happening seems low. The success of Llama and failure of Phi with $K = 8$ show that different LLMs have different sensitivity to the approximation. Figure 10a shows that that with a higher number of contexts $K > 32$ Phi is able to effectively learn, indicating Phi needs a higher computational budget. On the other hand, Llama is robust to the approximation, with most values performing similarly to Stochastic, except with the lowest values of $K$.

**Approximate Reduces Compute Needs.** We measure the reduction of tokens processed in Approximate compared to Stochastic throughout full ICRL runs. We approximate this measure by computing at each step the number of tokens required for a forward call and subtracting the number of tokens of the sequence with the longest common prefix processed in a previous step, as it would be possible to use the KV cache for all the tokens in the common prefix (assuming infinite memory). We find that Stochastic processes two orders of magnitude more tokens than Approximate. Table 7 provides numerical results for this analysis.

## C   Experimental Setup

We conduct experiments on various type of GPUs: 40GB A100, 80GB A100, 80GB H100, 48GB A6000. For experiments with 70B and 32B models, we use 4 80GB A100/H100 or 8 48GB A6000. For experiments with 14B models, we use 2 80GB A100/H100. For experiments with 7B or smaller models, we use 1 80GB A100/H100 or 2 48GB A6000 / 40GB A100. For efficient inference, we use *vllm* (Kwon et al., 2023).

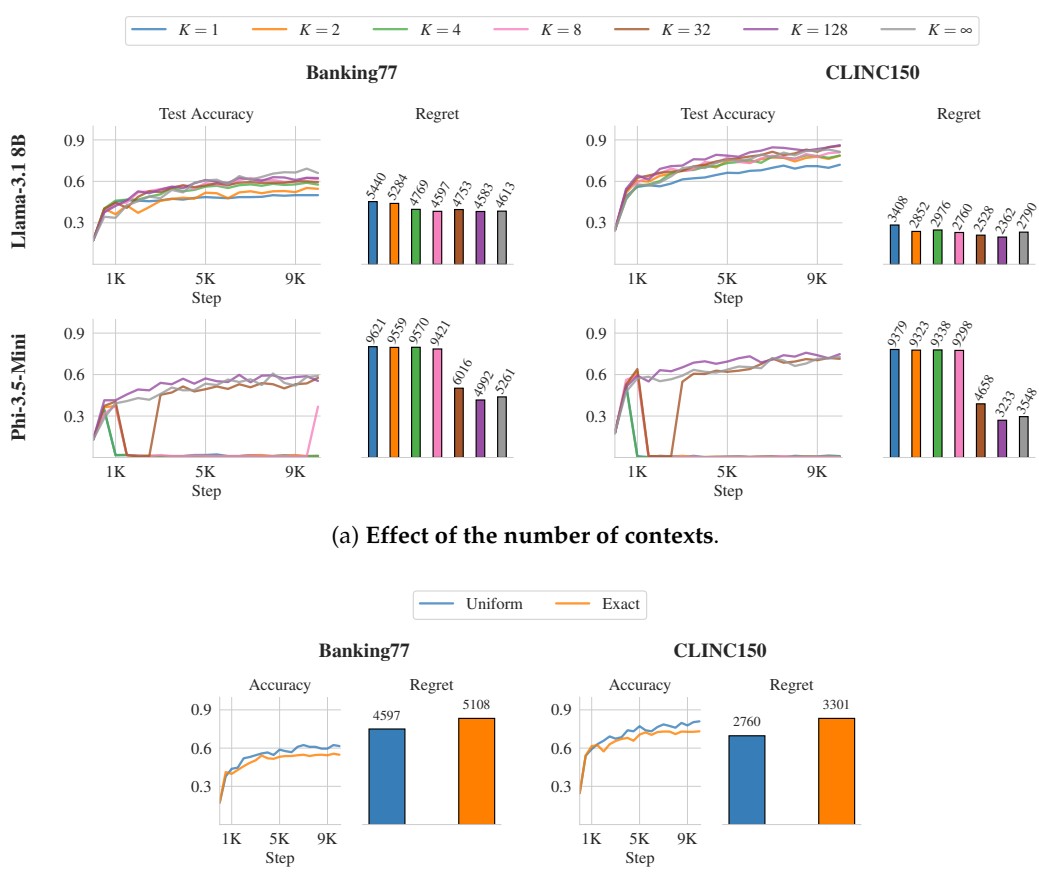

(a) **Effect of the number of contexts**.

(b) **Effect of the context sampling strategies**.

Figure 10: **Comparison of Approximate parameters**. (a) Effect of the number of contexts $K$. We report test accuracy for Llama and Phi. Phi proves more sensitive to this approximation. Generation degenerates for low $K$, while the model can learn for $K \geq 32$. Llama can learn with all $K$, although higher values perform better. (b) Comparison of exact and uniform sampling. We report test accuracy at the final step and regret for Llama. Uniform sampling strategy is consistently better.

## C.1 Prompt Design

We report prompt examples from ICL (Figure 11) and ICRL (Figure 12) experiments. We show the prompts for Llama as an example. In all cases, we show the prompts with two in-context examples.

## C.2 Context Windows and Episode Capacity

For each task and model combination, we conservatively estimate the maximum number of examples that could fit within the context window. This is done by including all observed examples in descending order of token count in the prompt, assuming the model consistently responds with the longest label and that the formatted reward message is at its maximum length. We perform this calculation using the maximum context window for all models. Additionally, for Llama, we repeat the process with context windows of 4,096 and 8,192 tokens specifically for the Banking77 and CLINC150 tasks. Table 1 reports episode capacity.

---

**Prompt example for ICL in Llama**

```
<|begin_of_text|><|start_header_id|>system<|end_header_id|>\n\nC
↪   utting Knowledge Date: December 2023\nToday Date: 26 Jul
↪   2024\n\nYou are an useful assistant. Answer the following
↪   questions.<|eot_id|><|start_header_id|>user<|end_header_id|>
↪   \n\nQuery: Tell me about the card
↪   PIN?<|eot_id|><|start_header_id|>assistant<|end_header_id|>\
↪   n\nIntent: get physical
↪   card<|eot_id|><|start_header_id|>user<|end_header_id|>\n\nQu
↪   ery: Is there a daily auto top-up
↪   limit?<|eot_id|><|start_header_id|>assistant<|end_header_id|
↪   >\n\nIntent: automatic top
↪   up<|eot_id|><|start_header_id|>user<|end_header_id|>\n\nQuer
↪   y: I got a message saying I made a withdrawal from the bank
↪   machine, but I did not.<|eot_id|><|start_header_id|>assistan
↪   t<|end_header_id|>\n\nIntent:
```

Figure 11: **An example of prompt of ICL for Llama.**

---

**Prompt example for ICRL in Llama**

```
<|begin_of_text|><|start_header_id|>system<|end_header_id|>\n\nC
↪   utting Knowledge Date: December 2023\nToday Date: 26 Jul
↪   2024\n\nYou are an useful assistant. Answer the following
↪   questions. Feedback will indicate if you answered correctly.
↪   You must answer correctly, using previous feedback to make
↪   better predictions.<|eot_id|><|start_header_id|>user<|end_he
↪   ader_id|>\n\nQuery: It declined my
↪   transfer.<|eot_id|><|start_header_id|>assistant<|end_header_
↪   id|>\n\nIntent: declined
↪   transfer<|eot_id|><|start_header_id|>user<|end_header_id|>\n
↪   \n'declined transfer' is the correct answer! Good
↪   job!\n\nQuery: Am I allowed to change my PIN anywhere?<|eot_
↪   id|><|start_header_id|>assistant<|end_header_id|>\n\nIntent:
↪   verify top
↪   up<|eot_id|><|start_header_id|>user<|end_header_id|>\n\nThe
↪   answer 'verify top up' is wrong! You can do better!\n\nQuery:
↪   If I'm getting my identity verified, what all do I need?<|eot
↪   _id|><|start_header_id|>assistant<|end_header_id|>\n\nIntent:
```

Figure 12: **An example of prompt of ICRL for Llama.**

## C.3 Datasets

We use 5 classification tasks (and the corresponding abstract-label variants) in our experiments:

- Banking77 (77 labels; Casanueva et al. (2020)). It involves 77 labels and aims to detect the intent of user queries in an economic context. For example, one label could be "*balance not updated after cheque or cash deposit*".

- CLINC150 (150 labels; Larson et al. (2019). It includes 150 labels, also focusing on intent classification. An example label is "*calendar update*". While the original dataset

Table 1: **Maximum number of episodes supported by model and task, given a specific context window**. We compute the maximum number of episodes supported by the context window of Llama, Phi, Qwen, and Gemini across all tasks, including 4k and 8k tokens for Llama, with Banking77 and CLINC150 only.

| | Phi | Llama | | | Qwen | Gemini |
|---|---|---|---|---|---|---|
| Task | 128k tokens | 4k tokens | 8k tokens | 128k tokens | 128k tokens | 1M tokens |
| **Banking77** | 1538 | 34 | 74 | 1673 | 1672 | - |
| **CLINC150** | 2241 | 60 | 126 | 2384 | 2184 | - |
| **NLU** | 2397 | - | - | 2425 | 2424 | - |
| **TREC** | 2848 | - | - | 2919 | 2896 | - |
| **TREC-fine** | 2584 | - | - | 2776 | 2755 | - |
| **Abs. Banking77** | - | - | - | 1924 | 1788 | 13007 |
| **Abs. CLINC150** | - | - | - | 2485 | 2270 | 19501 |
| **Abs. NLU** | - | - | - | 2475 | 2285 | 24603 |
| **Abs. TREC** | - | - | - | 2529 | 2308 | 5953 |
| **Abs. TREC-fine** | - | - | - | 2531 | 2308 | 5953 |

was designed to detect out-of-scope queries, we concentrate solely on classifying the 150 defined intents, excluding out-of-scope queries from our analysis.

- NLU (68 labels; Liu et al. (2021)). This dataset includes queries grouped in 68 unique categories for human-robot interaction in home domain (for example, one label is "*audio volume mute*".

- TREC and TREC-fine (respectively 6 and 50 labels; Li & Roth (2002); Hovy et al. (2001)). Both are question classification dataset where the goal is to classify the type of question. Each example contains both a fine label (that we use in TREC-fine), as "*entity vehicle*", and a coarse one (used in TREC), as "*entity*". TREC-fine includes 50 categories, while TREC groups them in only 6 categories.

All of these datasets are challenging because of the big number of different labels, and the sometimes subtle differences between labels. Moreover, in our setting we do not provide any information about the list of potential labels (except for the "With Exemplars" abstract-label experiments), challenging the model to either follow previously discovered labels or try to find new, more suitable ones (i.e., exploitation vs exploration – Sutton & Barto (2018)).

# D   Additional Results

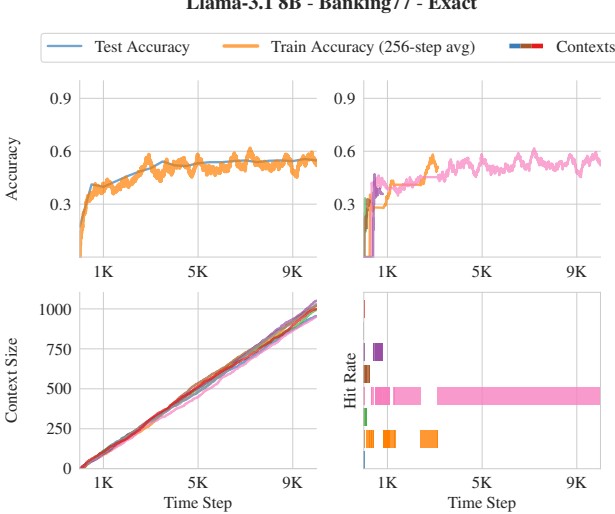

Figure 13: **Detailed visualization of Approximate for Llama, Banking77 with exact context sampling.** We report test accuracy (top left), a 256-step running average of the training accuracy (bottom left), the training accuracy of each context (top right), and the hit rate of each context (bottom right).

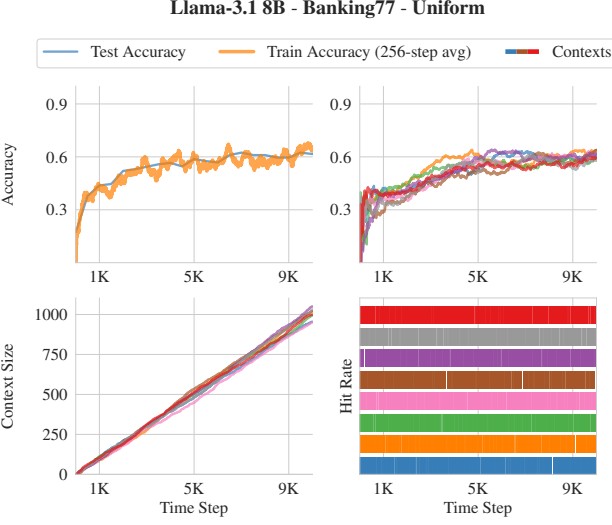

Figure 14: **Detailed visualization of Approximate for Llama, Banking77 with uniform context sampling.** We report test accuracy (top left), a 256-step running average of the training accuracy (bottom left), the training accuracy of each context (top right), and the hit rate of each context (bottom right).

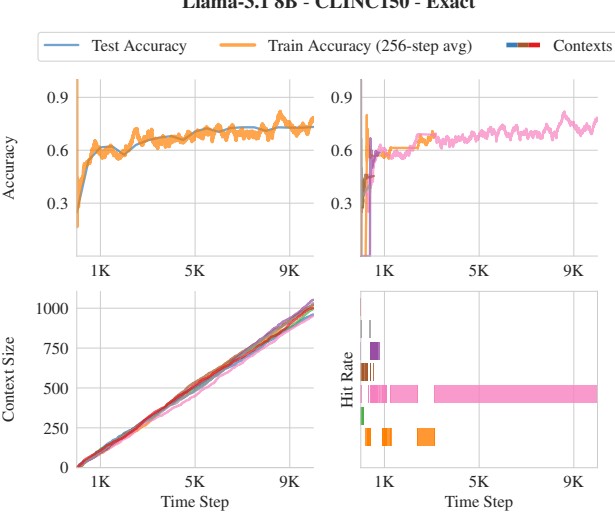

Figure 15: **Detailed visualization of Approximate for Llama, CLINC150 with exact context sampling.** We report test accuracy (top left), a 256-step running average of the training accuracy (bottom left), the training accuracy of each context (top right), and the hit rate of each context (bottom right).

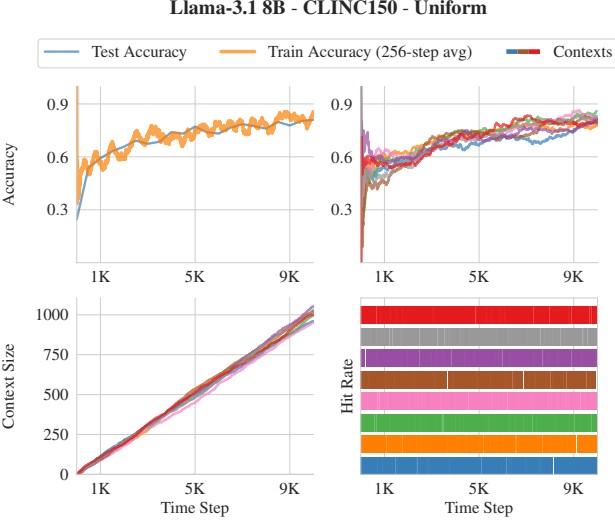

Figure 16: **Detailed visualization of Approximate for Llama, CLINC150 with uniform context sampling.** We report test accuracy (top left), a 256-step running average of the training accuracy (bottom left), the training accuracy of each context (top right), and the hit rate of each context (bottom right).

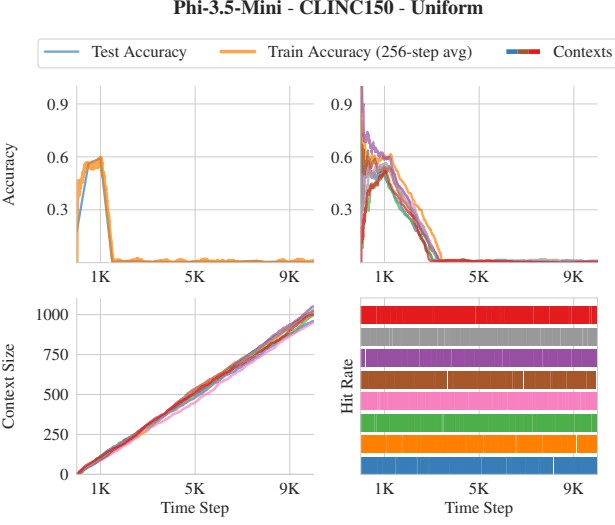

Figure 17: **Detailed visualization of Approximate for Phi, Banking77 with uniform context sampling.** We report test accuracy (top left), a 256-step running average of the training accuracy (bottom left), the training accuracy of each context (top right), and the hit rate of each context (bottom right).

Figure 18: **Detailed visualization of Approximate for Phi, CLINC150 with uniform context sampling.** We report test accuracy (top left), a 256-step running average of the training accuracy (bottom left), the training accuracy of each context (top right), and the hit rate of each context (bottom right).

Table 2: **Detailed Figures for Figure 2.** We report three key figures for each dataset and method: initial (zero-shot) accuracy, final (post-ICRL) accuracy, and regret (total mistakes). (a) contains the results for Llama, (b) for Qwen.

(a) **Llama**

| Dataset | Naive | | | Naive+ | | | Stochastic | | | Upper Bound |
|---|---|---|---|---|---|---|---|---|---|---|
| | 0-step Acc. | Final Acc. | Reg. | 0-step Acc. | Final Acc. | Reg. | 0-step Acc. | Final Acc. | Reg. | Acc. |
| **Banking77** | 0.172 | 0.104 | 8755 | 0.152 | 0.648 | 4358 | 0.172 | 0.660 | 4613 | 0.846 |
| **CLINC150** | 0.246 | 0.292 | 7126 | 0.092 | 0.836 | 2280 | 0.246 | 0.814 | 2790 | 0.944 |
| **NLU** | 0.338 | 0.322 | 6868 | 0.286 | 0.578 | 4486 | 0.338 | 0.706 | 3545 | 0.844 |
| **TREC** | 0.476 | 0.412 | 3692 | 0.326 | 0.744 | 2235 | 0.476 | 0.836 | 1183 | 0.872 |
| **TREC-fine** | 0.238 | 0.29 | 4390 | 0.070 | 0.610 | 2585 | 0.238 | 0.740 | 2183 | 0.760 |

(b) **Qwen**

| Dataset | Naive | | | Naive+ | | | Stochastic | | | Upper Bound |
|---|---|---|---|---|---|---|---|---|---|---|
| | 0-step Acc. | Final Acc. | Reg. | 0-step Acc. | Final Acc. | Reg. | 0-step Acc. | Final Acc. | Reg. | Acc. |
| **Banking77** | 0.064 | 0.174 | 8265 | 0.070 | 0.572 | 4433 | 0.062 | 0.722 | 4274 | 0.862 |
| **CLINC150** | 0.154 | 0.108 | 8691 | 0.138 | 0.802 | 2378 | 0.154 | 0.838 | 2913 | 0.960 |
| **NLU** | 0.216 | 0.230 | 7369 | 0.248 | 0.788 | 2967 | 0.208 | 0.748 | 3235 | 0.874 |
| **TREC** | 0.430 | 0.292 | 3883 | 0.372 | 0.664 | 2477 | 0.440 | 0.806 | 1181 | 0.880 |
| **TREC-fine** | 0.146 | 0.014 | 4734 | 0.122 | 0.536 | 3337 | 0.144 | 0.704 | 1920 | 0.812 |

Table 3: **Detailed Metrics for Figure 3.** We report three key metrics for each dataset and method: initial (zero-shot) accuracy, final (post-ICRL) accuracy, and regret (total mistakes). (a) shows results for Banking77, (b) for CLINC150.

(a) **Banking77**

| Reward | Naive | | | Stochastic | | |
|---|---|---|---|---|---|---|
| | 0-step Acc. | Final Acc. | Reg. | 0-step Acc. | Final Acc. | Reg. |
| **None** | 0.156 | 0.034 | 9426 | 0.172 | 0.308 | 7132 |
| **Only neg.** | 0.152 | 0.060 | 9331 | 0.172 | 0.214 | 7814 |
| **Only pos.** | 0.152 | 0.648 | 4358 | 0.172 | 0.660 | 4613 |
| **Both pos. and neg.** | 0.156 | 0.184 | 8624 | 0.172 | 0.458 | 5943 |
| **Noisy pos. and neg.** | 0.156 | 0.174 | 8713 | 0.172 | 0.394 | 6256 |
| **Inv. pos. and neg.** | 0.152 | 0.098 | 9234 | 0.172 | 0.282 | 7047 |

(b) **CLINC150**

| Reward | Naive | | | Stochastic | | |
|---|---|---|---|---|---|---|
| | 0-step Acc. | Final Acc. | Reg. | 0-step Acc. | Final Acc. | Reg. |
| **None** | 0.092 | 0.056 | 9214 | 0.246 | 0.388 | 6555 |
| **Only neg.** | 0.092 | 0.082 | 9010 | 0.246 | 0.208 | 7496 |
| **Only pos.** | 0.092 | 0.836 | 2280 | 0.246 | 0.814 | 2790 |
| **Both pos. and neg.** | 0.092 | 0.170 | 8280 | 0.246 | 0.582 | 4688 |
| **Noisy pos. and neg.** | 0.092 | 0.146 | 8454 | 0.246 | 0.586 | 4810 |
| **Inv. pos. and neg.** | 0.092 | 0.098 | 8995 | 0.246 | 0.448 | 5865 |

Table 4: **Detailed Figures for Figure 4.** We report three key figures for each dataset and method: initial (zero-shot) accuracy, final (post-ICRL) accuracy, and regret (total mistakes).

(a) Llama Accuracies

| Method | Abs. Banking77 | | Abs. CLINC150 | | Abs. NLU | | Abs. TREC | | Abs. TREC-fine | |
|---|---|---|---|---|---|---|---|---|---|---|
| | 0-step | Final | 0-step | Final | 0-step | Final | 0-step | Final | 0-step | Final |
| Naive | 0.020 | 0.018 | 0.002 | 0.000 | 0.010 | 0.028 | 0.030 | 0.086 | 0.034 | 0.030 |
| Naive+ (w/o Ex.) | 0.020 | 0.454 | 0.008 | 0.276 | 0.010 | 0.476 | 0.118 | 0.790 | 0.080 | 0.144 |
| Naive+ (w/ Ex.) | 0.024 | 0.178 | 0.000 | 0.590 | 0.056 | 0.656 | 0.238 | 0.792 | 0.006 | 0.196 |
| Stochastic (w/o Ex.) | 0.018 | 0.350 | 0.002 | 0.162 | 0.010 | 0.428 | 0.030 | 0.838 | 0.034 | 0.258 |
| Stochastic (w/ Ex.) | 0.030 | 0.584 | 0.006 | 0.650 | 0.162 | 0.722 | 0.238 | 0.840 | 0.008 | 0.036 |
| Up. Bound Acc. | | 0.780 | | 0.886 | | 0.750 | | 0.886 | | 0.612 |

(b) Llama Regrets

| Method | Abs. Banking77 | Abs. CLINC150 | Abs. NLU | Abs. TREC | Abs. TREC-fine |
|---|---|---|---|---|---|
| | Reg. | Reg. | Reg. | Reg. | Reg. |
| Naive | 9909 | 9928 | 9674 | 4032 | 4828 |
| Naive+ (w/o Ex.) | 7164 | 8507 | 6789 | 2046 | 4597 |
| Naive+ (w/ Ex.) | 4704 | 5727 | 4166 | 2214 | 4153 |
| Stochastic (w/o Ex.) | 8280 | 9437 | 7288 | 2244 | 4767 |
| Stochastic (w/ Ex.) | 4380 | 6276 | 3725 | 2424 | 4773 |

(c) Qwen Accuracies

| Method | Abs. Banking77 | | Abs. CLINC150 | | Abs. NLU | | Abs. TREC | | Abs. TREC-fine | |
|---|---|---|---|---|---|---|---|---|---|---|
| | 0-step | Final | 0-step | Final | 0-step | Final | 0-step | Final | 0-step | Final |
| Naive | 0.016 | 0.002 | 0.012 | 0.014 | 0.010 | 0.014 | 0.124 | 0.130 | 0.010 | 0.024 |
| Naive+ (w/o Ex.) | 0.014 | 0.442 | 0.008 | 0.204 | 0.014 | 0.622 | 0.136 | 0.526 | 0.010 | 0.146 |
| Naive+ (w/ Ex.) | 0.248 | 0.626 | 0.048 | 0.602 | 0.162 | 0.656 | 0.158 | 0.776 | 0.004 | 0.526 |
| Stochastic (w/o Ex.) | 0.016 | 0.208 | 0.012 | 0.134 | 0.010 | 0.564 | 0.124 | 0.886 | 0.010 | 0.532 |
| Stochastic (w/ Ex.) | 0.316 | 0.680 | 0.080 | 0.712 | 0.182 | 0.706 | 0.200 | 0.890 | 0.002 | 0.462 |
| Up. Bound Acc. | | 0.844 | | 0.902 | | 0.822 | | 0.892 | | 0.574 |

(d) Qwen Regrets

| Method | Abs. Banking77 | Abs. CLINC150 | Abs. NLU | Abs. TREC | Abs. TREC-fine |
|---|---|---|---|---|---|
| | Reg. | Reg. | Reg. | Reg. | Reg. |
| Naive | 9877 | 9892 | 9731 | 3876 | 4810 |
| Naive+ (w/o Ex.) | 7391 | 8824 | 6479 | 3304 | 3713 |
| Naive+ (w/ Ex.) | 4618 | 4481 | 3818 | 1564 | 3044 |
| Stochastic (w/o Ex.) | 8653 | 9449 | 6128 | 1591 | 3868 |
| Stochastic (w/ Ex.) | 4353 | 4385 | 3882 | 1520 | 4440 |

(e) Gemini Accuracies

| Method | Abs. Banking77 | | Abs. CLINC150 | | Abs. NLU | | Abs. TREC | | Abs. TREC-fine | |
|---|---|---|---|---|---|---|---|---|---|---|
| | 0-step | Final | 0-step | Final | 0-step | Final | 0-step | Final | 0-step | Final |
| Naive | 0.018 | 0.000 | 0.008 | 0.010 | 0.048 | 0.004 | 0.076 | 0.012 | 0.002 | 0.018 |
| Naive+ (w/o Ex.) | 0.014 | 0.174 | 0.002 | 0.144 | 0.044 | 0.218 | 0.104 | 0.900 | 0.006 | 0.130 |
| Naive+ (w/ Ex.) | 0.430 | 0.778 | 0.356 | 0.818 | 0.474 | 0.774 | 0.480 | 0.896 | 0.134 | 0.682 |
| Stochastic (w/o Ex.) | 0.014 | 0.214 | 0.008 | 0.060 | 0.050 | 0.290 | 0.072 | 0.876 | 0.002 | 0.060 |
| Stochastic (w/ Ex.) | 0.440 | 0.792 | 0.434 | 0.858 | 0.480 | 0.794 | 0.494 | 0.872 | 0.138 | 0.636 |
| Up. Bound Acc. | | 0.902 | | 0.964 | | 0.890 | | 0.946 | | 0.860 |

(f) Gemini Regrets

| Method | Abs. Banking77 | Abs. CLINC150 | Abs. NLU | Abs. TREC | Abs. TREC-fine |
|---|---|---|---|---|---|
| | Reg. | Reg. | Reg. | Reg. | Reg. |
| Naive | 9996 | 9929 | 9938 | 4841 | 4915 |
| Naive+ (w/o Ex.) | 8858 | 9264 | 7978 | 1308 | 3998 |
| Naive+ (w/ Ex.) | 2776 | 1711 | 2582 | 1127 | 2448 |
| Stochastic (w/o Ex.) | 8625 | 9710 | 8281 | 1505 | 4920 |
| Stochastic (w/ Ex.) | 3292 | 1984 | 2560 | 1439 | 3420 |

Table 5: **Detailed Metrics for Figure 5a.** We report three key metrics for each dataset and method: initial (zero-shot) accuracy, final (post-ICRL) accuracy, and regret (total mistakes). (a)–(g) show results for different sizes of Qwen2.5.

(a) **Qwen2.5 500M**

| Dataset | Naive+ 0-step | Naive+ Final | Naive+ Reg. | Stochastic 0-step | Stochastic Final | Stochastic Reg. |
|---|---|---|---|---|---|---|
| Banking77 | 0.082 | 0.284 | 7351 | 0.146 | 0.364 | 6874 |
| CLINC150 | 0.062 | 0.404 | 5959 | 0.118 | 0.552 | 5157 |
| NLU | 0.088 | 0.358 | 6746 | 0.146 | 0.544 | 6162 |
| TREC | 0.290 | 0.382 | 2260 | 0.318 | 0.514 | 2182 |
| TREC-fine | 0.032 | 0.154 | 3955 | 0.034 | 0.308 | 3961 |

(b) **Qwen2.5 1.5B**

| Dataset | Naive+ 0-step | Naive+ Final | Naive+ Reg. | Stochastic 0-step | Stochastic Final | Stochastic Reg. |
|---|---|---|---|---|---|---|
| Banking77 | 0.074 | 0.258 | 6069 | 0.092 | 0.524 | 5824 |
| CLINC150 | 0.066 | 0.482 | 4660 | 0.182 | 0.684 | 3945 |
| NLU | 0.242 | 0.302 | 5084 | 0.288 | 0.636 | 4619 |
| TREC | 0.274 | 0.302 | 2150 | 0.362 | 0.754 | 1835 |
| TREC-fine | 0.042 | 0.538 | 2979 | 0.074 | 0.680 | 2681 |

(c) **Qwen2.5 3B**

| Dataset | Naive+ 0-step | Naive+ Final | Naive+ Reg. | Stochastic 0-step | Stochastic Final | Stochastic Reg. |
|---|---|---|---|---|---|---|
| Banking77 | 0.082 | 0.532 | 4959 | 0.094 | 0.614 | 4852 |
| CLINC150 | 0.156 | 0.778 | 2370 | 0.148 | 0.812 | 2936 |
| NLU | 0.270 | 0.734 | 3355 | 0.266 | 0.772 | 2666 |
| TREC | 0.428 | 0.744 | 2482 | 0.506 | 0.730 | 1246 |
| TREC-fine | 0.098 | 0.520 | 3439 | 0.214 | 0.620 | 2131 |

(d) **Qwen2.5 7B**

| Dataset | Naive+ 0-step | Naive+ Final | Naive+ Reg. | Stochastic 0-step | Stochastic Final | Stochastic Reg. |
|---|---|---|---|---|---|---|
| Banking77 | 0.070 | 0.572 | 4433 | 0.062 | 0.722 | 4274 |
| CLINC150 | 0.138 | 0.802 | 2378 | 0.154 | 0.838 | 2913 |
| NLU | 0.248 | 0.788 | 2967 | 0.208 | 0.748 | 3235 |
| TREC | 0.372 | 0.664 | 2477 | 0.440 | 0.806 | 1181 |
| TREC-fine | 0.122 | 0.536 | 3337 | 0.144 | 0.704 | 1920 |

(e) **Qwen2.5 14B**

| Dataset | Naive+ 0-step | Naive+ Final | Naive+ Reg. | Stochastic 0-step | Stochastic Final | Stochastic Reg. |
|---|---|---|---|---|---|---|
| Banking77 | 0.126 | 0.506 | 4902 | 0.144 | 0.730 | 4136 |
| CLINC150 | 0.192 | 0.800 | 2505 | 0.248 | 0.882 | 2439 |
| NLU | 0.348 | 0.574 | 4198 | 0.376 | 0.748 | 3228 |
| TREC | 0.492 | 0.786 | 2105 | 0.556 | 0.904 | 919 |
| TREC-fine | 0.252 | 0.522 | 2989 | 0.322 | 0.734 | 1869 |

(f) **Qwen2.5 32B**

| Dataset | Naive+ 0-step | Naive+ Final | Naive+ Reg. | Stochastic 0-step | Stochastic Final | Stochastic Reg. |
|---|---|---|---|---|---|---|
| Banking77 | 0.132 | 0.518 | 5015 | 0.144 | 0.778 | 3733 |
| CLINC150 | 0.220 | 0.838 | 2164 | 0.280 | 0.884 | 2467 |
| NLU | 0.360 | 0.702 | 3588 | 0.380 | 0.768 | 2842 |
| TREC | 0.590 | 0.622 | 2185 | 0.646 | 0.920 | 770 |
| TREC-fine | 0.264 | 0.516 | 2800 | 0.354 | 0.662 | 1599 |

(g) **Qwen2.5 72B**

| Dataset | Naive+ 0-step | Naive+ Final | Naive+ Reg. | Stochastic 0-step | Stochastic Final | Stochastic Reg. |
|---|---|---|---|---|---|---|
| Banking77 | 0.186 | 0.592 | 4404 | 0.212 | 0.794 | 3512 |
| CLINC150 | 0.328 | 0.870 | 1714 | 0.360 | 0.906 | 1983 |
| NLU | 0.380 | 0.776 | 2520 | 0.438 | 0.794 | 2609 |
| TREC | 0.392 | 0.656 | 2277 | 0.416 | 0.898 | 782 |
| TREC-fine | 0.100 | 0.706 | 2398 | 0.170 | 0.764 | 1614 |

Table 6: **Detailed Stability Metric $\rho$ for Figure 5b.** Each cell contains the stability metric $\rho$ for the corresponding dataset and method. (a)–(g) show results for different sizes of Qwen2.5.

(a) **Qwen2.5 500M**

|  | Naive | Naive+ | Stochastic |
|---|---|---|---|
| **Banking77** | -0.343 | 0.491 | 0.710 |
| **CLINC150** | -0.459 | 0.634 | 0.877 |
| **NLU** | -0.025 | 0.045 | 0.868 |
| **TREC** | -0.500 | 0.264 | 0.700 |
| **TREC-fine** | -0.202 | 0.164 | 0.724 |

(b) **Qwen2.5 1.5B**

|  | Naive | Naive+ | Stochastic |
|---|---|---|---|
| **Banking77** | 0.209 | 0.427 | 0.875 |
| **CLINC150** | -0.369 | 0.278 | 0.949 |
| **NLU** | -0.260 | 0.088 | 0.851 |
| **TREC** | -0.500 | 0.064 | 0.501 |
| **TREC-fine** | 0.500 | 0.882 | 0.697 |

(c) **Qwen2.5 3B**

|  | Naive | Naive+ | Stochastic |
|---|---|---|---|
| **Banking77** | -0.069 | 0.710 | 0.932 |
| **CLINC150** | -0.652 | 0.748 | 0.956 |
| **NLU** | -0.857 | 0.711 | 0.925 |
| **TREC** | -0.500 | 0.882 | 0.773 |
| **TREC-fine** | 0.500 | 0.210 | 0.564 |

(d) **Qwen2.5 7B**

|  | Naive | Naive+ | Stochastic |
|---|---|---|---|
| **Banking77** | -0.653 | 0.694 | 0.989 |
| **CLINC150** | -0.794 | 0.679 | 0.985 |
| **NLU** | -0.570 | 0.661 | 0.963 |
| **TREC** | -0.230 | -0.134 | 0.314 |
| **TREC-fine** | -0.674 | 0.784 | 0.487 |

(e) **Qwen2.5 14B**

|  | Naive | Naive+ | Stochastic |
|---|---|---|---|
| **Banking77** | 0.394 | 0.548 | 0.964 |
| **CLINC150** | -0.730 | 0.956 | 0.981 |
| **NLU** | -0.494 | 0.337 | 0.859 |
| **TREC** | -0.790 | 0.927 | 0.441 |
| **TREC-fine** | -0.691 | 0.540 | 0.825 |

(f) **Qwen2.5 32B**

|  | Naive | Naive+ | Stochastic |
|---|---|---|---|
| **Banking77** | -0.218 | 0.684 | 0.972 |
| **CLINC150** | -0.419 | 0.939 | 0.989 |
| **NLU** | 0.035 | 0.887 | 0.945 |
| **TREC** | -0.791 | 0.074 | 0.822 |
| **TREC-fine** | -0.849 | 0.886 | 0.292 |

(g) **Qwen2.5 72B**

|  | Naive | Naive+ | Stochastic |
|---|---|---|---|
| **Banking77** | -0.456 | 0.894 | 0.965 |
| **CLINC150** | 0.159 | 0.854 | 0.986 |
| **NLU** | -0.633 | 0.461 | 0.910 |
| **TREC** | -0.500 | 0.936 | 0.765 |
| **TREC-fine** | -0.500 | 0.970 | 0.600 |

Table 7: **Tokens processed in Approximate compared to Stochastic throughout full ICRL runs.** Stochastic processes two orders of magnitude more tokens than Approximate.

|  | Phi | | | Llama | | |
|---|---|---|---|---|---|---|
| **Task** | Expl. | Approx. | Ratio | Expl. | Approx. | Ratio |
| **Banking77** | 87,369,607 | 510,786 | 171 | 102,282,989 | 539,367 | 190 |
| **CLINC150** | 105,545,002 | 398,677 | 265 | 122,455,599 | 440,019 | 278 |
| **NLU** | 89,894,548 | 409,680 | 219 | 114,517,653 | 433,254 | 264 |
| **TREC** | 29,306,971 | 212,855 | 138 | 34,509,170 | 229,046 | 151 |
| **TREC-fine** | 20,658,980 | 222,955 | 93 | 25,522,358 | 234,884 | 109 |

