# OpenReview forum: "LLMs Are In-Context Bandit Reinforcement Learners"
_colmweb.org/COLM/2025/Conference — COLM 2025_

### Official Review · Reviewer_dGPZ · 2025-05-12

**Rating:** 4
**Confidence:** 3
**Ethics Flag:** 1

**Summary:**

This paper starts from the premise that LLMs are generally quite good at ICL, and shows that LLMs are also amenable to a contextual bandit version of in-context RL. There are two versions, a Naïve version and a stochastic ICRL version

**Questions To Authors:**

-	Are prompts and context really interchangeable (L20) or is the latter more precisely a type or part of the former?
-	Why are some references incomplete (e.g., Bertsch et al (2024) or improperly formatted/capitalized?
-	To what extent is it a limitation that single-step bandits are the exclusive focus of this paper?
-	Woiuld it be possible to correct ‘negligeble’ to ‘negligible’? L259

**Reasons To Accept:**

-	A good number of NLP tasks are included (i.e., CLINIC, NLU, TREC)
-	To the extent to which ICL and RL overlap, it is good that they are both addressed, although this is not the first work to do so.
-	The empirical methodology and variants of tasks (if not the models) is positive, and there is some analysis on scaling although additional forms of ablation, error analysis, or deeper analysis generally would have been possible.

**Reasons To Reject:**

-	The reward system from oracle knowledge may appear to be too simplistic or naïve. Some form of evaluation that includes uncertainty, types of noise, etc, should be considered.
-	A deeper theoretical rationale and a clearer differentiation from previous work in ICRL is suggested.

---

> ### Author Response · Authors · 2025-05-30
>
> We thank the reviewer for their comments\!
>
> **“To the extent to which ICL and RL overlap, it is good that they are both addressed, although this is not the first work to do so.” … “A deeper theoretical rationale and a clearer differentiation from previous work in ICRL is suggested.”**
>
> To the best of our knowledge, ours is the first work to show that off-the-shelf LLMs can learn NLP tasks (where they initially fail) through ICRL, and to propose methods that avoid the failure modes of the most naive versions of ICRL. Krishnamurthy et al. 2024, which is indeed prior work (as we discuss in the related work), focused on toy bandit tasks, considered only bandit setting (no context), and largely showed a negative result. Nie et al. 2024 (concurrent work to ours) studied contextual bandits, but relied on intensive fine-tuning of models to get ICRL-like behavior (and even then with explicit aggregation of statistics external to the model). If there is work that we missed, we will be happy to discuss it, and will appreciate pointers to it.
>
> **“The reward system from oracle knowledge may appear to be too simplistic or naïve. Some form of evaluation that includes uncertainty, types of noise, etc, should be considered.”**
>
> We include studies with noisy rewards (in Figure 3), showing that the ICRL behavior we identify is robust. The popular recent direction of RL for LLMs through verifiable rewards \[1, 2\] also makes use of binary rewards with exact-match with ground-truth answers, for example for math tasks, which is very similar to our reward function. While simple, this is an effective reward paradigm, and widely adopted.
>
> **“To what extent is it a limitation that single-step bandits are the exclusive focus of this paper?”**
>
> We agree that there are interesting follow up work to be done on multi-step RL. However, single-step bandits are the de-facto formulation of all recent successful RL methods for LLMs, that reach SOTA through RL training \[2\], as, just like for us, multiple tokens are considered a single action and accordingly rewarded (this includes RLHF). Given the recent success of outcome-oriented bandit-like RL methods for LLMs, it is clear that there is a lot of value in focusing on the single-step bandit setting, and we leave the multi-step RL setting for future work.
>
> \[1\] Lambert et al., “Tulu 3: Pushing Frontiers in Open Language Model Post-Training”
>
> \[2\] DeepSeek-AI, “Deepseek-R1: Incentivizing reasoning capability in LLMS via reinforcement learning”

---

### Official Review · Reviewer_WrPe · 2025-05-13

**Rating:** 7
**Confidence:** 5
**Ethics Flag:** 1

**Summary:**

The authors introduce a protocol for in-context learning using a contextual bandit RL setup. The authors extend the common in-context learning prompting setup for LLMs (including examples in the prompt) into a setting where the signal/output is obtained from an external source.
The approach effectively makes prompts stateful by appending (input, response, reward). In the experiments, the bandit setting is implemented by taking single labels from multi-label classification benchmarks. Prompts do not contain the actual labels, only rewards derived from the ground-truth datasets.

In practice the method requires prompt modification to work -- the authors observe that positive triples only works better than putting full histories in prompts, and implement an episode sampling protocol which leads to more robust evaluation results. The final method is referred to as "Stochastic ICRL".
Although the approach is quite simple, and many questions remain to be answered, the paper is well written and the authors conduct thorough evaluation and ablation.

**Questions To Authors:**

- 159: zero-short --> zero-shot
- Note -- the "ICRL" acronym is close to "ICLR", and is misspelled in line 47 -- this creates extra cognitive load for ML folks when reading, I immediately think of the conference for a split-second every time i read that
- because evaluation is stateful, ordering of dataset matters, suggest shuffling and making sure results hold

**Reasons To Accept:**

- extends ICL to signal based rewards with stateful prompts, a setting that will be interesting for many LLM practitioners.
- well written paper with good contextualisation in current research streams
- thorough evaluation and ablation

**Reasons To Reject:**

- the positive only and stochastic sampling results are not examined in enough depth to result in satisfactory takeaways for future work.
- especially why models can deal with negative samples in stochastic paradigm but not in the base setting is a counter-intuitive result which requires more study, the risk of bugs or artifacts of the setup here is high in my opinion
- the key takeaways are close to "LLMs do better at new tasks when prompts contain lots of clear information about the task", which is already obvious

---

> ### Author Response · Authors · 2025-05-30
>
> We thank the reviewer for their comments\!
>
> Thank you for identifying the typos. Apologies. We will fix them in the next version.
>
> “**the key takeaways are close to "LLMs do better at new tasks when prompts contain lots of clear information about the task", which is already obvious”**
>
> Our main research questions are not around the amount of information in the prompt and relation to task performance. The key research questions (and takeaways) are about how the information in the prompt (i.e, infill) is obtained. The key takeaway from our work is that gold labels are not needed to improve model performance, and model outputs with an external verifier/reward function are sufficient.  In other words, weak supervision from a verifier is enough for prompt self-improvement, compared to the standard strong supervision from gold labels. This is quite helpful for NLP practitioners, as it does not rely on gold labels.
>
> **“especially why models can deal with negative samples in stochastic paradigm but not in the base setting is a counter-intuitive result which requires more study, the risk of bugs or artifacts of the setup here is high in my opinion”**
>
> There are several potential explanations. First, Naive’s context grows much faster. Therefore, it can become very biased towards what happens in the beginning of the ICRL learning process, where the majority of the episodes receive negative rewards . The model may be inclined to repeat mistakes, because mistakes have a massive presence in the beginning. This is less of a problem for Stochastic, because (in our main experiments) only 10% of all samples are used every time, so the model can see fewer mistakes (many early episodes are dropped through the stochastic sampling)and has more time to explore. Also, Naive’s context always contains the same episodes. On the other hand, stochastic has a different subset of episodes every time, which can introduce additional variance/randomness (and less “bias”) in the model behavior, benefiting exploration. We discuss this to some degree in lines 230-234, but will improve the discussion given the points above.
>
> **“the positive only and stochastic sampling results are not examined in enough depth to result in satisfactory takeaways for future work.”**
>
> We are happy to conduct further analysis. Are there concrete suggestions? Or maybe concrete questions or nullifiable alternative hypotheses? We can’t be sure that we will get them in time for the discussion period, but can definitely provide further analyses for the next version of the paper. Thanks.

---

> > ### Comment · Reviewer_WrPe · 2025-06-09
> >
> > Thanks for your response, I'm maintaining my good score for this paper, but I do have some comments
> >
> > >Are there concrete suggestions? Or maybe concrete questions or nullifiable alternative hypotheses?
> >
> > one experiment that could be easy to run would be to update the prompt to make the model "aware" of the experimental setup, and explicitly say that earlier items may be incorrect, or to explicitly assign a weight to each example in addition to outputting the classification result. I.e. the prompt says
> > ```
> > <your task description ...>
> > Your output should be based upon the information contained in the examples, therefore, also assign a scalar weight between [0,1] to each feedback item, indicating how much that example impacted your decision. Remember that the examples are ordered, and because they are generated by a stochastic process that is exploring a large space, earlier examples are likely to contain negative rewards ...
> > ```
> >
> > or simply say "ignore negative rewards" in the prompt
> > if these simple experiments don't improve performance of the naive setting, I think there are likely deeper bugs in the setup, but as the review states, I think this is a well-written paper with useful insights, and I would like to see it appear at COLM

---

> > > ### Author Response · Authors · 2025-06-09
> > >
> > > Our prompt already says something like this:
> > > ```
> > > <|begin_of_text|><|start_header_id|>system<|end_header_id|>\n\nCutting Knowledge Date: December 2023\nToday Date: 26 Jul 2024\n\nYou are an useful assistant. Answer the following questions. Feedback will indicate if you answered correctly. You must answer correctly, using previous feedback to make better predictions.<|eot_id|>
> > > ```
> > > It's not exactly what you write, but it's in the same spirit. We searched the space of prompts, and it didn't really change things in a meaningful way.
> > >
> > > We understand the concern about our results (with all the great stuff that LLMs achieve, it's at times hard to accept negative results), but we are certain we have no bug (and we are releasing everything). Critically, our results align very well with [Krishnamurty et al. 2024](https://arxiv.org/abs/2403.15371), and the concurrently released work of [Nie et al.](https://arxiv.org/abs/2410.06238). So this gives us a very robust verification of the phenomena we are observing. Both of these address different benchmarks. Kirshnamurthy does bandit learning (no context), and reports negative results. Nie studies toy contextual bandit problems, and takes a different approach of fine-tuning an LLM to replicate parts of the UCB algorithm. We elicit ICRL in a more native way.

---

### Official Review · Reviewer_cGRh · 2025-05-13

**Rating:** 6
**Confidence:** 3
**Ethics Flag:** 1

**Summary:**

This paper explores a contextual bandit version of in-context reinforcement learning (ICRL), using rewards instead of supervised data. The authors consider Naive, Naive+, and stochastic ICRL, and show that LLMs can learn in-context for both semantically meaningful labels and abstract labels from their predictions and positive rewards, and larger models perform better.

**Reasons To Accept:**

This paper explores in-context reinforcement learning and the impact of positive and negative rewards. Solid experiments consider both semantically meaningful labels and abstract labels. While related in spirit to retrospective and feedback-based prompting methods, this work frames the problem more explicitly as reinforcement learning in a contextual bandit setting, and shows that LLMs learn best from positive reward alone.

**Reasons To Reject:**

The reward design is relatively simple, and it's unclear how generalizable it is. The conclusion—that using only positive rewards leads to the best performance—feels quite close to a supervised learning setup.

---

> ### Author Response · Authors · 2025-05-30
>
> We thank the reviewer for their comments\!
>
> **“The reward design is relatively simple, and it's unclear how generalizable it is”**
>
> We agree that the reward design is simple. That said, it’s a common design that has been shown to be effective again and again – it is the same reward design structure in the RL from verifiable reward setting (RLVR) \[1, 2\]. In particular, in these settings, it has been shown that having a simple reward that verifies if an answer is correct or not can be extremely helpful in training a policy \[1, 2\].
>
> **“The conclusion—that using only positive rewards leads to the best performance—feels quite close to a supervised learning setup.”**
>
> The similarity is in form only. Critically, the labels in ICRL are generated by the model and are not provided with the data. Instead, ICRL relies on a reward function (i.e., verifier). So, while in form the prompt text looks similar, the fundamentals of the data are completely different.
>
> \[1\] Lambert et al., “Tulu 3: Pushing Frontiers in Open Language Model Post-Training”
>
> \[2\] DeepSeek-AI, “Deepseek-R1: Incentivizing reasoning capability in LLMS via reinforcement learning”

---

### Decision · Program_Chairs · 2025-07-06

**Decision:**

Accept

**Comment:**

This paper presents an empirical study of a contextual bandit version of ICRL (in-context reinforcement learning) using LLMs, showing that models improve performance primarily through positive feedback signals. The experiments are thorough and cover a diverse set of NLP tasks, demonstrating the efficacy of using rewards within prompts to guide LLM behavior. Reviewers have recognized the clarity, practical value, and experimental rigor of the paper.

However, a key conceptual concern emerges regarding the distinction between traditional ICL and the particular instantiation of ICRL studied here. Consider a scenario where an LLM initially produces random answers and subsequently receives both positive and negative feedback. If we assume the LLMs are good enough to effectively ignore negative feedback, then the scenario effectively collapses back to standard ICL, a domain in which LLMs are known to excel.

This raises an important conceptual ambiguity: what precisely is the difference between an LLM that can do standard ICL effectively, and one that performs the version of ICRL described in this paper? Without explicit comparisons against classical contextual-bandit baselines using metrics such as regret, the claim that "LLMs can do in-context reinforcement learning" remains weak, since, under the above reasoning, any LLM proficient in standard ICL might appear to be a "no-regret learner" by default.

Additionally, although deep theoretical grounding is not a strict necessity, the paper would substantially benefit from thorough contextualization against many theoretical works (Lin et al. 2024, Park et al. 2024), highlighting critical discussions around the precise conceptual and practical definitions of ICRL.

Balancing these strengths and weaknesses, especially the critical conceptual overlap with standard ICL and the absence of classical regret-based baselines, I recommend acceptance while strongly encouraging the authors to revise the work based on the above discussion points.